# Neural Pose Representation Learning for Generating and Transferring Non-Rigid Object Poses

**Seungwoo Yoo    Juil Koo    Kyeongmin Yeo    Minhyuk Sung**
KAIST
{dreamy1534,63days,aaaaa,mhsung}@kaist.ac.kr

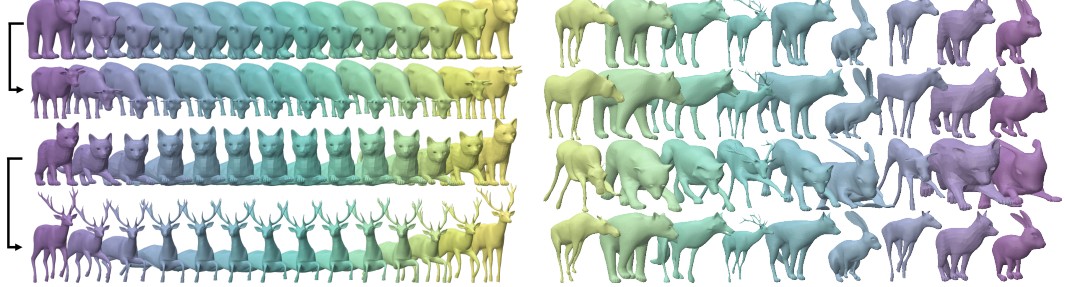

Figure 1: Results of motion sequence transfer (left) and shape variation generation (right) using the proposed neural pose representation. On the left, poses from source shapes (first and third rows) are transferred to target shapes (second and fourth rows), preserving intricate details like horns and antlers. On the right, new poses sampled from a cascaded diffusion model, trained with shape variations of the bunny (last column), are transferred to other animal shapes.

## Abstract

We propose a novel method for learning representations of poses for 3D deformable objects, which specializes in 1) disentangling pose information from the object's identity, 2) facilitating the learning of pose variations, and 3) transferring pose information to other object identities. Based on these properties, our method enables the generation of 3D deformable objects with diversity in both identities and poses, using variations of a single object. It does not require explicit shape parameterization such as skeletons or joints, point-level or shape-level correspondence supervision, or variations of the target object for pose transfer. We first design the pose extractor to represent the pose as a keypoint-based hybrid representation and the pose applier to learn an implicit deformation field. To better distill pose information from the object's geometry, we propose the implicit pose applier to output an intrinsic mesh property, the face Jacobian. Once the extracted pose information is transferred to the target object, the pose applier is fine-tuned in a self-supervised manner to better describe the target object's shapes with pose variations. The extracted poses are also used to train a cascaded diffusion model to enable the generation of novel poses. Our experiments with the DeformThings4D and Human datasets demonstrate state-of-the-art performance in pose transfer and the ability to generate diverse deformed shapes with various objects and poses.

## 1 Introduction

The recent great success of generative models [17, 39, 40, 38] has been made possible not only due to advances in techniques but also due to the enormous scale of data that has become available, such as LAION [35] for 2D image generation. For 3D data, the scale has been rapidly increasing, as exemplified by ObjaverseXL [10]. However, it is still far from sufficient to cover all possible 3D shapes, particularly deformable, non-rigid 3D shapes such as humans, animals, and characters. The challenge with deformable 3D shapes is especially pronounced due to the diversity in *both*

38th Conference on Neural Information Processing Systems (NeurIPS 2024).

the *identities* and *poses* of the objects. Additionally, for a new 3D character created by a designer, information about possible variations of the creature does not even exist.

To remedy the requirement of a large-scale dataset for 3D deformable shape generation, we aim to answer the following question: Given variations of a *single* deformable object with its different poses, how can we effectively learn the pose variations while factoring out the object's identity and also make the pose information applicable to other objects? For instance, when we have a variety of poses of a bear (Fig. 1 left, first row), our objective is to learn the space of poses without entangling them with the geometric characteristics of the bears. Also, we aim to enable a sample from this space to be applied to a new object, such as a bull, to generate a new shape (Fig. 1 left, second row). We believe that such a technique, effectively separating pose from the object's identity and enabling the transfer of poses to other identities, can significantly reduce the need for collecting large-scale datasets covering the diversity of both object identities and poses. This approach can even enable creating variations of a new creature without having seen any possible poses of that specific object.

Transferring poses from one object to another has been extensively studied in computer graphics and vision, with most methods requiring target shape supervision [42, 5, 51, 4, 12] or predefined pose parameterization [14, 9, 22, 43, 3, 50, 33, 11, 45, 20, 25, 46, 8]. Without such additional supervision, our key idea for extracting identity-agnostic pose information and learning their variations is to introduce a novel pose representation along with associated encoding and decoding techniques. For this, we consider the following three desiderata:

1. **Pose Disentanglement**: The representation should effectively represent the pose only without resembling the source object's identity when applied to the other object.
2. **Compactness**: The representation should be compact enough to effectively learn its variation using a generative model, such as a diffusion model.
3. **Transferability**: The encoded pose information should be applicable to new target objects.

As a representation that meets the aforementioned criteria, we propose an autoencoding framework and a latent diffusion model with three core components. Firstly, we design a *pose extractor* and a *pose applier* to encode an **implicit deformation field** with a **keypoint-based hybrid representation**, comprising 100 keypoints in the space, each associated with a latent feature. Learning the deformation field enables *disentangling* the pose information from the object's identity, while the keypoint-based representation *compactly* encodes it and makes it *transferable* to other objects. However, simply learning the deformation as a new position of the vertex is not sufficient to properly adapt the source object's pose information to others. Hence, secondly, we propose predicting an **intrinsic property** of the deformed mesh, **Jacobian fields** [54, 26, 41, 2], which can successfully apply the pose while preserving the identity of the target shape. To better preserve the target's identity while applying the pose variation from the source, thirdly, we propose a **per-identity refinement step** that fine-tunes the decoder in a *self-supervised* way to adapt to the variations of target shapes, with poses transferred from the source object. Thanks to the compact hybrid representation of pose, a pose generative model can also be effectively learned using **cascaded diffusion models** [18, 21], enabling the generation of varying poses of an object with an arbitrary identity different from the source object.

In our experiments, we compare our framework against state-of-the-art techniques for pose transfer on animals (Sec. 4.2) and humans (Sec. 4.3). Both qualitative and quantitative analyses underscore the key design factors of our framework, demonstrating its efficiency in capturing identity-agnostic poses from exemplars and its superior performance compared to existing methods. Additionally, we extend the proposed representation to the task of unconditional generation of shape variations. Our representation serves as a compact encoding of poses that can be generated using diffusion models (Sec. 4.6) and subsequently transferred to other shapes. This approach facilitates the generation of various shapes, particularly in categories where exemplar collection is challenging.

## 2   Related Work

Due to space constraints, we focus on reviewing the literature on non-rigid shape pose transfer, including methods that operate without parameterizations, those that rely on predefined parameterizations, and recent learning-based techniques that derive parameterizations from data.

**Parameterization-Free Pose Transfer.**   Early works [42, 49, 5, 51] focused on leveraging *pointwise* correspondences between source and target shapes. A seminal work by Sumner and Popović [42] transfers per-triangle affine transforms applied to the target shape by solving an optimization problem. A follow-up work by Ben-Chen *et al.* [5] transfers deformation gradients by approximating

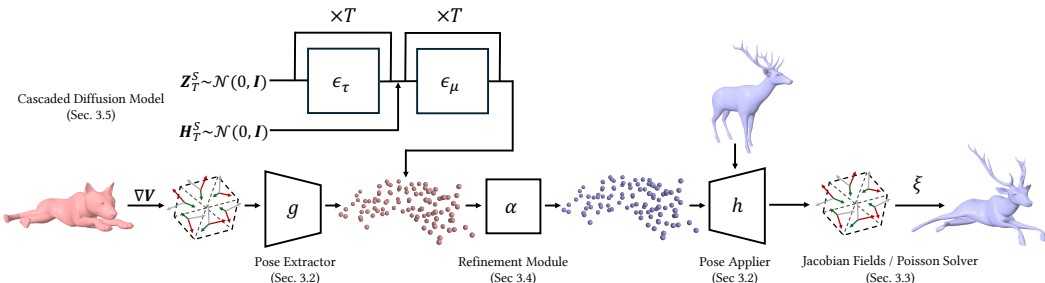

Figure 2: **Method overview.** Our framework extracts keypoint-based hybrid pose representations from Jacobian fields. These fields are mapped by the pose extractor $g$ and mapped back by the pose applier $h$. The pose applier, conditioned on the extracted pose, acts as an implicit deformation field for various shapes, including those unseen during training. A refinement module $\alpha$, positioned between $g$ and $h$, is trained in a self-supervised manner, leveraging the target's template shape. The compactness of our latent representations facilitates the training of a diffusion model, enabling diverse pose variations through generative modeling in the latent space.

source deformations using harmonic bases. On the other hand, a technique proposed by Baran *et al.* [4] instead employs *pose-wise* correspondences by learning shape spaces from given pairs of poses shared across the source and target identities. The poses are transferred by blending existing exemplars. While these techniques require point-wise or pose-wise correspondence supervision, our method does not require such supervision during training or inference.

**Skeleton- or Joint-Based Pose Transfer.** Another line of work utilizes handcrafted skeletons, which facilitate pose transfer via motion retargeting [14]. This approach has been extended by incorporating physical constraints [9, 22, 43, 3] or generalizing the framework to arbitrary objects [50, 33]. Several learning-based methods [11, 45, 20, 25, 46] have also been proposed to predict joint transformations involved in forward kinematics from examples. Recently, Chen *et al.* [8] proposed a framework that does not require skeletons during test time by predicting keypoints at joints. The method is trained to predict both relative transformations between corresponding keypoints in two distinct kinematic trees and skinning weights. However, the tasks of rigging and skinning are labor-intensive, and different characters and creatures often require distinct rigs with varying topologies. Liao *et al.* [24] notably presented a representation that comprises character-agnostic deformation parts and a semi-supervised network predicting skinning weights that link each vertex to these deformation parts, although its performance hinges on accurate skinning weight prediction. In this work, we design a more versatile framework that is applicable to various shapes and provides better performance.

**Pose Transfer via Learned Parameterization.** To bypass the need for correspondence or parameterization supervision, learning-based approaches [12, 53, 47, 57, 7, 2, 24, 48, 36, 37] explore alternative parameterizations *learned* from exemplars. Yifan *et al.* [53] propose to predict source and target cages and their offsets simultaneously, although their method still requires manual landmark annotations. Gao *et al.* [12] introduce a VAE-GAN framework that takes *unpaired* source and target shape sets, each containing its own set of pose variations. The network is trained without direct pose-wise correspondences between samples from these sets, instead enforcing cycle consistency between latent representations. Although this work relaxes the requirement for correspondence supervision, it still requires pose variations for *both* the source and target identities and individual training for each new source-target pair. Numerous works [47, 7, 57, 2] lift the requirement for gathering variations of target shapes by disentangling identities from poses, enforcing cycle consistency [57], or adapting conditional normalization layers [47] from image style transfer [30]. Notably, Aigerman *et al.* [2] train a network that regresses Jacobian fields from SMPL [28] pose parameters. The vertex coordinates are computed by solving Poisson's equation [54], effectively preserving the shapes' local details. Wang *et al.* [48] also train a neural implicit function and retrieve a shape latent from a template mesh of an unseen identity via autodecoding [29]. This method models local deformations through a coordinate-based network that learns continuous deformation fields. However, such methods struggle to generalize to unseen identities due to their reliance on *global* latent embeddings encoding shapes. We propose a representation that not only disentangles poses from identities but also allows for implicit queries using the surface points, thereby improving generalization to new identities.

## 3 Method

### 3.1 Problem Definition

Consider a *source template mesh* $\overline{\mathcal{M}}^S = (\overline{\mathbf{V}}^S, \mathbf{F}^S)$, given as a 2-manifold triangular mesh. The mesh comprises vertices $\overline{\mathbf{V}}^S$ and faces $\mathbf{F}^S$. Suppose there exist $N$ variations of the source template mesh, $\{\mathcal{M}_1^S, \dots, \mathcal{M}_N^S\}$, where each $\mathcal{M}_i^S = (\mathbf{V}_i^S, \mathbf{F}^S)$ is constructed with a different *pose*, altering the vertex positions while sharing the same mesh connectivity $\mathbf{F}^S$.

Assume a *target template mesh* $\overline{\mathcal{M}}^T = (\overline{\mathbf{V}}^T, \mathbf{F}^T)$ is given without any information about its variations or existing pose parameterization (*e.g.,*, skeletons or joints). Our goal is to define functions $g$ and $h$ that can *transfer* the pose variations from the source meshes to the target template mesh. For each variation of the source shape $\mathcal{M}_i^S$, its corresponding mesh $\mathcal{M}_i^T$ for the target is obtained as:

$$\mathcal{M}_i^T = (h(g(\mathcal{M}_i^S), \overline{\mathcal{M}}^T), \mathbf{F}^T), \quad \text{for } i = 1, 2, \cdots, N. \tag{1}$$

Specifically, we design $g$ as a *pose extractor* that embeds a source object mesh $\mathcal{M}_i^S$ into a *pose latent representation* $\mathcal{Z}_i^S = g(\mathcal{M}_i^S)$. This representation disentangles the pose information from the object's identity in $\mathcal{M}_i^S$ and facilitates transferring the pose to the target template mesh $\overline{\mathcal{M}}^T$. Given this pose representation, the *pose applier* $h$ then applies the pose to $\overline{\mathcal{M}}^T$, yielding the corresponding variation of the target object $\mathcal{M}_i^T = h(\mathcal{Z}_i^S, \overline{\mathcal{M}}^T)$. Note that our method is not limited to transferring the pose of a *given* variation of the source object to the target mesh but can also apply a pose *generated* by a diffusion model to the target mesh. In Sec. 3.5, we explain how a diffusion model can be trained with the latent pose representation extracted from source object variations. In the following, we first describe the key design factors of the functions $g$ and $h$ to tackle the problem.

### 3.2 Keypoint-Based Hybrid Pose Representation

To encode a source shape $\mathcal{M}^S$ into a latent representation $\mathcal{Z}^S$, we consider its vertices $\mathbf{V}^S$ as its geometric representation and use them as input to the pose extractor $g$, which is designed as a sequence of Point Transformer [56, 44] layers. These layers integrate vector attention mechanisms [55] with progressive downsampling of input point clouds. The output of the pose extractor $g$ is a set of unordered tuples $\mathcal{Z}^S = \left\{ (\mathbf{z}_k^S, \mathbf{h}_k^S) \right\}_{k=1}^K$ where $\mathbf{z}_k^S \in \mathbb{R}^3$ represents a 3D coordinate of a keypoint subsampled from $\mathbf{V}^S$ via farthest point sampling (FPS) and $\mathbf{h}_k^S$ is a learned feature associated with $\mathbf{z}_k^S$. This set $\mathcal{Z}^S$ forms a sparse point cloud of keypoints in 3D space, augmented with latent features. We set $K = 100$ in our experiments.

This keypoint-based hybrid representation, visualized in Fig. 2, is designed to exclusively transfer pose information from the source to the target while preventing leakage of the source shape's identity characteristics. Since the keypoints $\{\mathbf{z}_k^S\}$ are sampled from $\mathbf{V}^S$ using FPS, they effectively capture the overall pose structure of $\mathbf{V}^S$ while also supporting geometric queries with the vertices of a new mesh. This property is essential during the decoding phase, where the pose applier $h$ predicts the pose-applied mesh from the input template as an *implicit deformation field*.

The pose applier $h$ is implemented with a neural network that takes the 3D coordinates of a vertex from the input template mesh as a query, along with the hybrid pose latent representation $\mathcal{Z}$, and outputs the new position of the vertex in the pose-applied deformed mesh. Note that $h$ indicates a function that collectively maps all vertices in the input template mesh to their new positions using the network. Like the pose extractor $g$, the implicit deformation network is also parameterized as Point Transformer layers [56]. It integrates the pose information encoded in $\mathcal{Z}^S$ by combining vector attention mechanisms with nearest neighbor queries to aggregate features of the keypoints $\mathbf{z}_k^S$ around each query point. The aggregated features are then decoded by an MLP to predict the vertex coordinates of the deformed shape. (This is a base network, and we also introduce a *better* way to design the implicit deformation network in Sec. 3.3.)

Given only the variations of the source object $\{\mathcal{M}_1^S, \dots, \mathcal{M}_N^S\}$, we jointly learn the functions $g$ and $h$ by reconstructing the variations of the source object as a deformation of its template:

$$\mathcal{L}_V = \|\mathbf{V}_i^S - h(g(\mathcal{M}_i^S), \overline{\mathcal{M}}^S)\|^2. \tag{2}$$

While $g$ and $h$ are trained using the known variations of the source object $\mathcal{M}^S$, the latent representation $\mathcal{Z}^S = g(\mathcal{M}^S)$, when queried and decoded with the target template mesh, effectively transfers the

pose extracted by $g$ from $\mathcal{M}^S$. However, we also observe that $g$ and $h$, when trained using the loss $\mathcal{L}_V$, often result in geometry with noticeable imperfections and noise on the surfaces. To address this, we explore an alternative representation of a mesh that better captures and preserves geometric details, which will be discussed in the following section.

### 3.3 Representing Shapes as Jacobian Fields

In this work, we advocate employing the differential properties of surfaces as dual representations of a mesh. Of particular interest are *Jacobian fields*, a gradient-domain representation noted for its efficacy in preserving local geometric details during deformations [2], while ensuring that the resulting surfaces maintain smoothness [13].

Given a mesh $\mathcal{M} = (\mathbf{V}, \mathbf{F})$, a Jacobian field $\mathbf{J}$ represents the spatial derivative of a scalar-valued function $\phi : \mathcal{M} \to \mathbb{R}$ defined over the surface. We discretize $\phi$ as $\phi_{\mathbf{V}} \in \mathbb{R}^{|\mathbf{V}|}$, sampling its value at each vertex $\mathbf{v}$ of the vertex set $\mathbf{V}$. The spatial derivative of $\phi$ at each triangle $\mathbf{f} \in \mathbf{F}$ is computed as $\boldsymbol{\nabla}_{\mathbf{f}} \phi_{\mathbf{V}}$ using the per-triangle gradient operator $\boldsymbol{\nabla}_{\mathbf{f}}$. Given that each dimension of vertex coordinates $\mathbf{V}$ is such a function, we compute its spatial gradient at each triangle $\mathbf{f}$ as $\mathbf{J}_{\mathbf{f}} = \boldsymbol{\nabla}_{\mathbf{f}} \mathbf{V}$. Iterating this process for all triangles yields the Jacobian field $\mathbf{J} = \{\mathbf{J}_{\mathbf{f}} | \mathbf{f} \in \mathbf{F}\}$.

To recover $\mathbf{V}$ from a given Jacobian field $\mathbf{J}$, we solve a least-squares problem, referred to as Poisson's equation:

$$\mathbf{V}^* = \underset{\mathbf{V}}{\operatorname{argmin}} \|\mathbf{L}\mathbf{V} - \boldsymbol{\nabla}^T \mathcal{A}\mathbf{J}\|^2, \tag{3}$$

where $\mathbf{L} \in \mathbb{R}^{|\mathbf{V}| \times |\mathbf{V}|}$ is the cotangent Laplacian of $\mathcal{M}$, $\boldsymbol{\nabla}$ is the stack of gradient operators defined at each $\mathbf{f} \in \mathbf{F}$, and $\mathcal{A} \in \mathbb{R}^{3|\mathbf{F}| \times 3|\mathbf{F}|}$ is the diagonal mass matrix, respectively. Since the rank of $\mathbf{L}$ is at most $|\mathbf{V}| - 1$, we can obtain the solution by fixing a single point, which is equivalent to eliminating one row of the system in Eqn. 3. Since $\mathbf{L}$ in Eqn. 3 remains constant for a given shape $\mathcal{M}$, we can prefactorize the matrix (*e.g.,* using Cholesky decomposition) and quickly solve the system for different Jacobian fields $\mathbf{J}$'s. Furthermore, the upstream gradients can be propagated through the solver since it involves only matrix multiplications [2].

Employing Jacobian fields as shape representations, we now modify the implicit deformation network described in Sec. 3.2 to take *face center coordinates* as input instead of vertex coordinates and to output a new *face Jacobian* for a query face instead of a new vertex position. This results in decomposing $h$ into two functions $h = (\xi \circ h')$, where $h'$ is a function that collectively maps all the faces in the input mesh to the new Jacobian, and $\xi$ is a differentiable Poisson solver layer. Both $g$ and $h'$ are then trained by optimizing the following loss:

$$\mathcal{L}_J = \|\mathbf{V}_i^S - \xi(h'(g(\mathcal{M}_i^S), \overline{\mathcal{M}}^S))\|^2. \tag{4}$$

### 3.4 Per-Identity Refinement using Geometric Losses

While the latent pose representation $\mathcal{Z}$ learned by $g$ and $h$ exhibits promising generalization capabilities in transferring poses, the quality of the transferred shapes can be further improved by incorporating a trainable, identity-specific refinement module into our system. This module is trained in a *self-supervised* manner with the set of pose-applied target meshes. Similarly to techniques for personalized image generation [19, 52], we introduce a shallow network $\alpha$ between $g$ and $h$, optimizing its parameters while keeping the rest of the pipeline frozen.

The optimization of $\alpha$ is driven by geometric losses, aiming to minimize the geometric discrepancies in terms of the object's identity between the target template mesh $\overline{\mathcal{M}}^T$ and the pose-transferred meshes $\mathcal{M}_i^T$. In particular, we first extract poses $\{\mathcal{Z}_1^S, \ldots, \mathcal{Z}_N^S\}$ corresponding to the known shapes $\{\mathcal{M}_1^S, \ldots, \mathcal{M}_N^S\}$ of the source object. A transformer-based network $\alpha$, which maps a latent representation $\mathcal{Z}^S$ to $\mathcal{Z}^{S'}$, is plugged in between the pose extractor $g$ and the pose applier $h$:

$$\mathbf{V}_i^T = h(\alpha(\mathcal{Z}_i^S), \overline{\mathcal{M}}^T). \tag{5}$$

The parameters of $\alpha$ are updated by optimizing the following loss function:

$$\mathcal{L}_{\text{ref}} = \lambda_{\text{lap}} \mathcal{L}_{\text{lap}}(\mathbf{V}_i^T, \overline{\mathbf{V}}^T) + \lambda_{\text{edge}} \mathcal{L}_{\text{edge}}(\mathbf{V}_i^T, \overline{\mathbf{V}}^T) + \lambda_{\text{reg}}(\sum_k \|\mathbf{z}_k^S - \mathbf{z}_k^{S'}\|^2 + \|\mathbf{h}_k^S - \mathbf{h}_k^{S'}\|^2), \tag{6}$$

where $\mathcal{L}_{\text{lap}}(\cdot)$ is the mesh Laplacian loss [27], $\mathcal{L}_{\text{edge}}(\cdot)$ is the edge length preservation loss [24], and $\lambda_{\text{lap}}$, $\lambda_{\text{edge}}$, and $\lambda_{\text{reg}}$ are the weights of the loss terms. The definitions of the losses are provided in the appendix. Note that this refinement step leverages only the originally provided template shape $\overline{\mathcal{M}}^T$ and does not require its given variations or any other additional supervision.

## 3.5   Learning Latent Diffusion via Cascaded Training

The use of the keypoint-based hybrid representation discussed in Sec. 3.2 offers a compact latent space suitable for generative modeling using diffusion models [39, 17, 40, 38]. Unlike the Jacobian fields with dimensionality $|\mathbf{F}| \times 9$, the keypoints and their feature vectors that comprise the pose representation $\mathcal{Z}^S$ lie in significantly lower dimensional space, facilitating generative modeling with latent diffusion models [34].

We employ a cascaded diffusion framework [18, 21] to separately capture the layouts of keypoints and the associated feature vectors. Given a set $\{\mathcal{Z}_1^S, \ldots, \mathcal{Z}_N^S\}$ of $N$ latent embeddings extracted from the known source shape variations $\{\mathcal{M}_1^S, \ldots, \mathcal{M}_N^S\}$, we first learn the distribution over $\mathbf{Z}^S = \{\mathbf{z}_k^S\}_{k=1}^K$. To handle unordered sets with small cardinality, we employ a transformer-based network to facilitate interactions between each element within the noise prediction network $\boldsymbol{\epsilon}_\tau(\mathbf{Z}_t^S, t)$ where $t$ is a diffusion timestep and $\mathbf{Z}_t^S$ is a noisy 3D point cloud obtained by perturbing a clean keypoint set $\mathbf{Z}_0^S \left(= \mathbf{Z}^S\right)$ via forward diffusion process [17]. We train the network by optimizing the denoising loss:

$$\mathcal{L}_{\mathbf{Z}^S} = \mathbb{E}_{\mathbf{Z}^S, \boldsymbol{\epsilon} \sim \mathcal{N}(\mathbf{0}, \mathbf{I}), t \sim \mathcal{U}(0,1)} \left[ \|\boldsymbol{\epsilon} - \boldsymbol{\epsilon}_\tau(\mathbf{Z}_t^S, t)\|^2 \right]. \tag{7}$$

Likewise, the distribution of the set of latent features $\mathbf{H}^S = \left\{\mathbf{h}_k^S\right\}_{k=1}^K$ is modeled as a conditional diffusion model $\boldsymbol{\epsilon}_\mu$, which takes $\mathbf{Z}^S$ as an additional input to capture the correlation between $\mathbf{Z}^S$ and $\mathbf{H}^S$. The network is trained using the same denoising loss:

$$\mathcal{L}_{\mathbf{H}^S} = \mathbb{E}_{\mathbf{H}^S, \boldsymbol{\epsilon} \sim \mathcal{N}(\mathbf{0}, \mathbf{I}), t \sim \mathcal{U}(0,1)} \left[ \|\boldsymbol{\epsilon} - \boldsymbol{\epsilon}_\mu(\mathbf{H}_t^S, \mathbf{Z}^S, t)\|^2 \right]. \tag{8}$$

Once trained, the models can sample new pose representations through the reverse diffusion steps [17]:

$$\mathbf{Z}_{t-1}^S = \frac{1}{\sqrt{\alpha_t}} \left( \mathbf{Z}_t^S - \frac{\beta_t}{\sqrt{1 - \bar{\alpha}_t}} \boldsymbol{\epsilon}_\tau \left(\mathbf{Z}_t^S, t\right) \right), \tag{9}$$

$$\mathbf{H}_{t-1}^S = \frac{1}{\sqrt{\alpha_t}} \left( \mathbf{H}_t^S - \frac{\beta_t}{\sqrt{1 - \bar{\alpha}_t}} \boldsymbol{\epsilon}_\mu \left(\mathbf{H}_t^S, \mathbf{Z}_0^S, t\right) \right), \tag{10}$$

where $\alpha_t$, $\bar{\alpha}_t$, and $\beta_t$ are the diffusion process coefficients.

# 4   Experiments

## 4.1   Experiment Setup

**Datasets.** In our experiments, we consider animal and human shapes that are widely used in various applications. For the animal shapes, we utilize animation sequences from the DeformingThings4D-Animals dataset [23]. Specifically, we extract 300 meshes from the animation sequences of each of 9 different animal identities, spanning diverse species such as bears, rabbits, dogs, and moose. For humanoids, we use SMPL [28, 31], which facilitates easy generation of synthetic data for both training and testing. We sample 300 random pose parameters from VPoser [31] to generate variations of an unclothed human figure using the *default* body shape parameters, which are used to train our networks. For testing, we keep the pose parameters constant and sample 40 different body shapes from the parametric space covered by the unit Gaussian. This produces 40 different identities, each in 300 poses. The generated meshes serve as the ground truth for pose transfer. To assess the generalization capability to unusual identities that deviate significantly from the default body shape, we increase the standard deviation to 2.5 when sampling SMPL body parameters for 30 of the 40 identities. Additionally, we collect 9 stylized character meshes from the Mixamo dataset [1] to test the generalizability of different methods. For diffusion model training, the extracted keypoints and their associated features from the given set of source meshes are used as the training data for our cascaded diffusion model, which is trained separately for each identity.

**Baselines.** To assess the performance of pose transfer, we compare our method against Aigerman et al. [2] (NJF), Liao et al. [24] (SPT), Wang et al. [48] (ZPT), and various modifications of our

framework. For NJF [2], we use the official code from the *Morphing Humans* experiment, employing a PointNet [32] encoder to map input shapes to global latents, and we train the model on our datasets. For SPT [24], we use the official code and pretrained model on humanoid shapes. Since a pretrained model for animal shapes is not provided, the comparison with SPT on the DeformingThings4D-Animals dataset is omitted. For ZPT [48], the official implementation is not provided, so we implemented the model based on the description in the paper. In our ablation study, we explore different variations of our method to assess their impact on performance, including: (1) using vertex coordinates as shape representations (as described in Sec. 3.2), and (2) omitting the per-identity refinement module (Sec. 3.4). All models (except for SPT [24], for which we employ a pretrained model) are trained for *each* shape identity.

**Evaluation Metrics.** For pose transfer, when the corresponding shapes of the same pose are given for both source and target shapes, in the SMPL case, we measure accuracy using Point-wise Mesh Euclidean Distance (PMD) [57, 47], following our baselines[24, 48]. Note that this measurement cannot be applied in the DeformingThings4D-Animals case since pose-wise correspondences are not provided. For both pose transfer and shape generation (via pose generation), we measure the visual plausibility of the output meshes using FID [16], KID [6], and ResNet classification accuracy with images rendered from four viewpoints (front, back, left, and right) without texture. For the latter, we train a ResNet-18 [15] network using 10,800 images rendered from the four viewpoints of all ground truth shape variations of each animal.

## 4.2 Pose Transfer on DeformingThings4D-Animals

We begin our experiments by transferring pose variations across different animals, a challenging task that necessitates strong generalization capabilities due to the diverse shapes of the animals involved.

| | DeformingThings4D-Animals [23] | | | SMPL [28] | | | |
| | FID ↓ ($\times 10^{-2}$) | KID ↓ ($\times 10^{-2}$) | ResNet Acc. ↑ (%) | PMD ↓ ($\times 10^{-3}$) | FID ↓ ($\times 10^{-2}$) | KID ↓ ($\times 10^{-2}$) | ResNet Acc. ↑ (%) |
|---|---|---|---|---|---|---|---|
| NJF [2] | 11.33 | 5.71 | 64.43 | 2.55 | 1.57 | 0.82 | 70.93 |
| SPT [24] | - | - | - | 0.28 | 0.83 | 0.43 | 75.38 |
| ZPT [48] | 19.88 | 11.09 | 48.15 | 1.28 | 0.77 | 0.45 | 69.88 |
| Ours | **1.11** | **0.42** | **78.72** | **0.13** | **0.30** | **0.19** | **79.09** |

Table 1: Quantitative results on the experiments using the DeformingThings4D-Animals dataset [23] and the human shape dataset populated using SMPL [28].

We summarize the quantitative metrics in Tab. 1 (left). Our method outperforms NJF [2] and ZPT [48], both of which use global latent codes to encode shapes, while ours uses a keypoint-based hybrid representation. Note that SPT [24] is not compared in this experiment since the pretrained model is not provided for animal shapes. Qualitative results are also shown in Fig. 3, demonstrating the transfer of poses from a source mesh $\mathcal{M}^S$ (second and seventh column, *red*) to a target template mesh $\overline{\mathcal{M}}^T$ (first and sixth column, *blue*). Both NJF [2] and ZPT [48] introduce significant distortions to the results and often fail to properly align the pose extracted from the source to the target. Our method, on the other hand, effectively transfers poses to the targets while preserving local geometric details. More results can be found in the appendix.

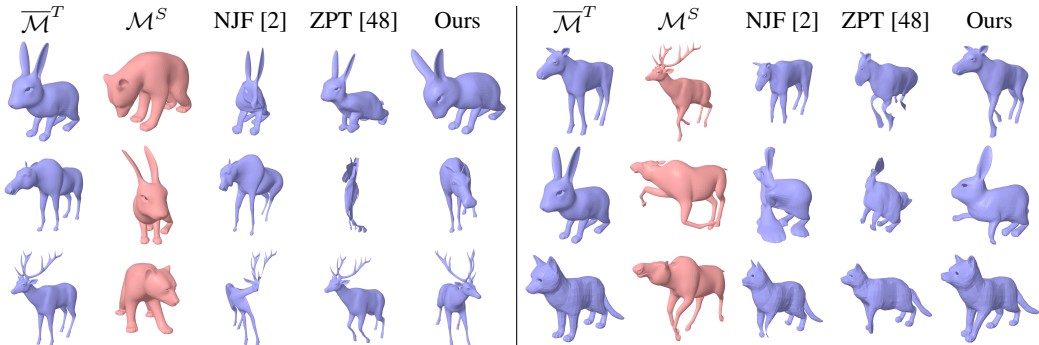

Figure 3: Qualitative results of transferring poses of the source meshes $\mathcal{M}^S$'s (*red*) in the DeformingThings4D animals [23] to target templates $\overline{\mathcal{M}}^T$'s (*blue*). Best viewed when zoomed in.

### 4.3 Pose Transfer on SMPL and Mixamo

We further test our method and baselines using humanoid shapes ranging from SMPL to stylized characters from the Mixamo [1] dataset. While we employ the parametric body shape and pose model of SMPL [28, 31], it is important to note that this is only for evaluation purposes; our method does not assume any parametric representations, such as skeletons, for either training or inference.

Tab. 1 (right) summarizes the evaluation metrics measured across the 40 target shapes. Notably, our method achieves lower PMD than SPT [24], which is trained on a large-scale dataset consisting of diverse characters and poses, while ours is trained using only 300 pose variations of the default human body. This is further illustrated in the qualitative results in Fig. 4, where we demonstrate pose transfer from source meshes (*red*) to target template meshes (*blue*) not seen during training. As shown, the shapes transferred by our method accurately match the overall poses. Our method benefits from combining a keypoint-based hybrid representation with Jacobian fields, outperforming the baselines in preserving local details, especially in areas with intricate geometric features such as the hands. See the zoomed-in views in Fig. 4. More results can be found in the appendix.

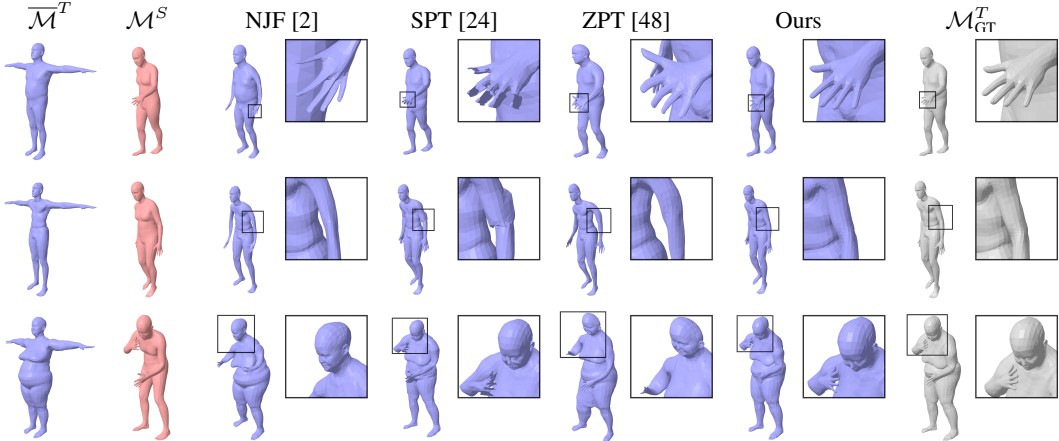

Figure 4: Qualitative results of transferring poses of the default human meshes $\mathcal{M}^S$'s (*red*) to different target template meshes $\overline{\mathcal{M}}^T$'s (*blue*). The ground truth targets $\mathcal{M}^T_{\text{GT}}$'s (*grey*) are displayed for reference. Best viewed when zoomed in.

Furthermore, we apply our model to a more challenging setup involving stylized characters. In Fig. 5, we present qualitative results using shapes from the Mixamo [1] dataset. Despite being trained on a single, unclothed SMPL body shape, our method generalizes well to stylized humanoid characters with detailed geometry (first row) and even to a character missing one arm (second row).

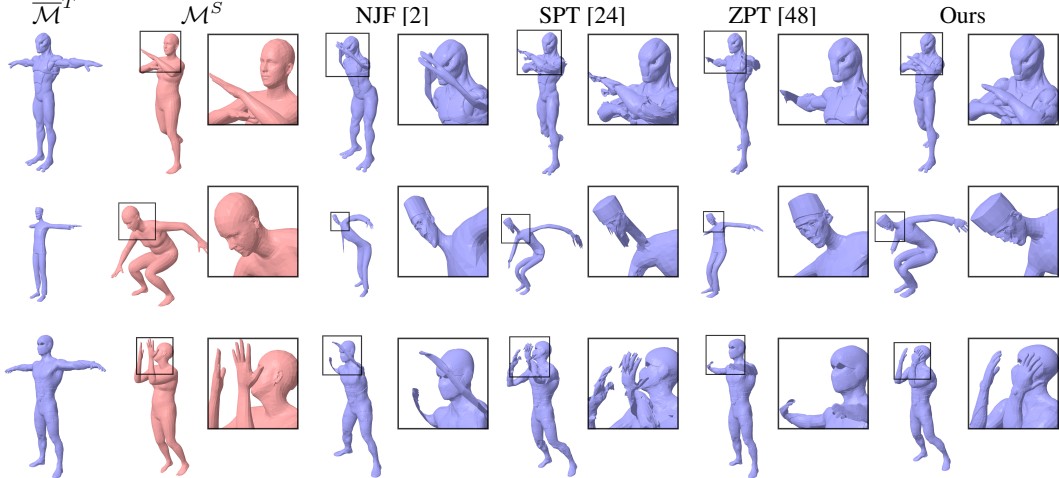

Figure 5: Qualitative results of transferring poses of the default human meshes $\mathcal{M}^S$'s (*red*) to target template meshes $\overline{\mathcal{M}}^T$'s (*blue*) of Mixamo characters [1]. Best viewed when zoomed in.

## 4.4 Ablation Study

Our framework design is further validated by comparisons against different variations of our framework, as listed in Sec. 4.1, in the pose transfer experiment on animal shapes discussed in Sec. 4.2. Tab. 2 (left) summarizes the image plausibility metrics measured using the results from our internal baselines. Qualitative results are presented in Fig. 6. Our method, which extracts pose representations from Jacobian fields and leverages the per-identity refinement module, achieves the best performance among all the variations.

| Jacobian (Sec. 3.3) | Refinement (Sec. 3.4) | Poses from $\mathcal{M}_i^S$ | | | Generated Poses (Sec. 3.5) | | |
|---|---|---|---|---|---|---|---|
| | | FID $\downarrow$ ($\times 10^{-2}$) | KID $\downarrow$ ($\times 10^{-2}$) | ResNet Acc. $\uparrow$ (%) | FID $\downarrow$ ($\times 10^{-2}$) | KID $\downarrow$ ($\times 10^{-2}$) | ResNet Acc. $\uparrow$ (%) |
| ✗ | ✗ | 3.52 | 2.13 | 55.67 | 9.52 | 4.69 | 44.34 |
| ✓ | ✗ | 1.17 | 0.47 | 75.13 | 4.40 | 2.39 | 75.82 |
| ✓ | ✓ | **1.11** | **0.42** | **78.72** | **4.22** | **2.24** | **78.81** |

Table 2: Ablation study using the poses from the source shapes in DeformingThings4D-Animals [23] dataset (left) and the poses generated from our cascaded diffusion model.

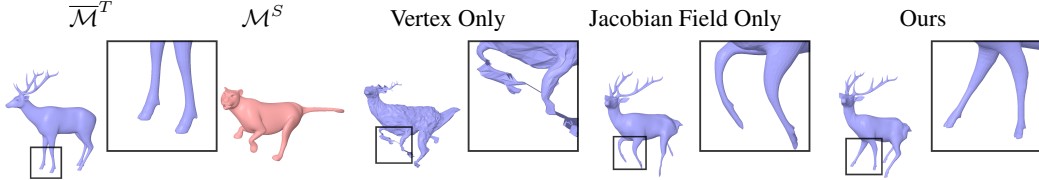

Figure 6: Qualitative results from the ablation study where a pose of the source shape $\mathcal{M}^S$ (*red*) in the DeformingThings4D-Animals [23] is transferred to the target template shape $\overline{\mathcal{M}}^T$ (*blue*).

## 4.5 Sensitivity to Number of Keypoints

We examine the sensitivity of our method to the number of keypoints by testing different variants of our framework while varying the number of keypoints extracted by the pose extractor to 50, 25, and 10, respectively. These variants are trained using the same SMPL [28] human body shapes and animal shapes from the DeformingThings4D-Animals [23] dataset. Our per-identity refinement stage (Sec. 3.4) is omitted to focus exclusively on the impact of keypoint counts on performance. Tab. 3 summarizes FID and PMD measured using the DeformingThings4D-Animals and SMPL dataset, respectively. We showcase qualitative results in Fig. 7 and Fig. 8. As reflected in both quantitative and qualitative results, reducing the number of keypoints does not significantly affect pose transfer accuracy.

| | DeformingThings4D-Animals [23] | | | | | SMPL [28] | | | |
|---|---|---|---|---|---|---|---|---|---|
| Method | Ours–10 | Ours-25 | Ours-50 | **Ours-100** | Method | Ours–10 | Ours-25 | Ours-50 | **Ours-100** |
| FID ($\times 10^{-2}$) | 1.25 | 0.87 | 0.83 | **0.72** | PMD ($\times 10^{-3}$) | 0.20 | 0.17 | 0.17 | **0.13** |

Table 3: Quantitative results from the variants of our framework trained to extract different number of keypoints. Ours-$N$ denotes a variant of our network trained to extract $N$ keypoints.

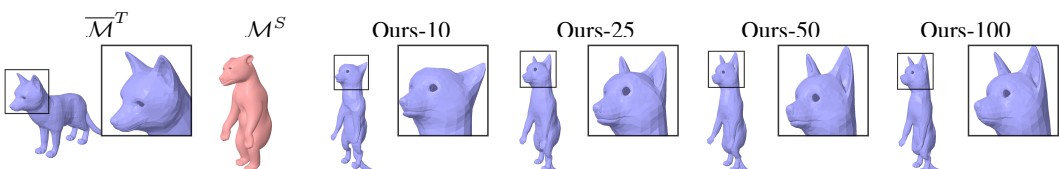

Figure 7: Qualitative results of transferring a pose of the source shape $\mathcal{M}^S$ (*red*) in the DeformingThings4D-Animals [23] to the target template shape $\overline{\mathcal{M}}^T$ (*blue*) using variants of our framework (Ours-$N$), trained to extract $N$ keypoints.

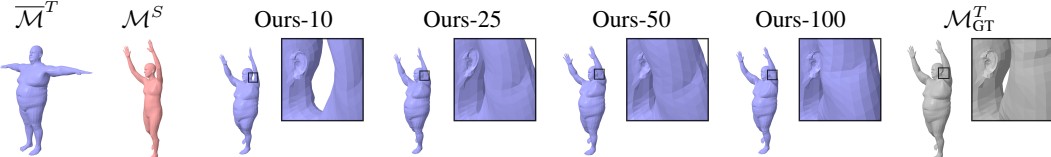

$\overline{\mathcal{M}}^T \qquad \mathcal{M}^S \qquad$ Ours-10 $\qquad$ Ours-25 $\qquad$ Ours-50 $\qquad$ Ours-100 $\qquad \mathcal{M}^T_{\mathrm{GT}}$

Figure 8: Qualitative results of transferring a pose of the default human mesh $\mathcal{M}^S$ (*red*) to the target template mesh $\overline{\mathcal{M}}^T$ (*blue*) using variants of our framework (Ours-$N$), trained to extract $N$ keypoints.

## 4.6 Pose Variation Generation Using Diffusion Models

We evaluate the generation capabilities of our diffusion models trained using different pose representations. Since no existing generative model can learn pose representations transferable across various shapes, we focus on analyzing the impact of using Jacobian fields on generation quality. We use shapes obtained by applying 300 generated poses to both $\overline{\mathcal{M}}^S$'s (*red*) and various $\overline{\mathcal{M}}^T$'s (*blue*). The quantitative and qualitative results are summarized in Tab. 2 (right) and Fig. 9, respectively. While the poses are generated using the diffusion model, our model still achieves ResNet classification accuracy comparable to the pose transfer experiment (Tab. 2, left). This tendency is also reflected in the qualitative results shown in Fig. 9. These results validate that the latent space learned from variations of Jacobian fields is more suitable for generating high-quality shape and pose variations compared to the one based on vertices. More results can be found in the appendix.

$\mathcal{M}^S$ (Vertex) $\quad$ $\mathcal{M}^S$ (Ours) $\quad$ $\mathcal{M}^T$ (Vertex) $\quad$ $\mathcal{M}^T$ (Ours) $\quad$ $\mathcal{M}^S$ (Vertex) $\quad$ $\mathcal{M}^S$ (Ours) $\quad$ $\mathcal{M}^T$ (Vertex) $\quad$ $\mathcal{M}^T$ (Ours)

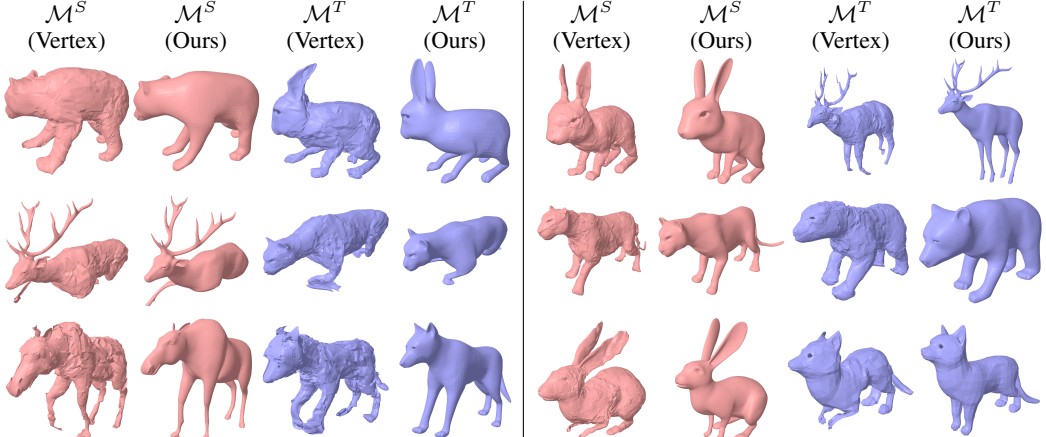

Figure 9: Pose variation generation results. Each row illustrates the outcomes of applying a generated pose to a source template mesh $\overline{\mathcal{M}}^S$ (*red*) and a target template mesh $\overline{\mathcal{M}}^T$ (*blue*).

## 5 Conclusion

We have presented a method for learning a novel neural representation of the pose of non-rigid 3D shapes, which facilitates: 1) the disentanglement of pose and object identity, 2) the training of a generative model due to its compactness, and 3) the transfer of poses to other objects' meshes. In our experiments, we demonstrated the state-of-the-art performance of our method in pose transfer, as well as its ability to generate diverse shapes by applying the generated poses to different identities.

**Limitations.** Our method leverages differential operators to compute the Jacobian field of the given template mesh, requiring additional preprocessing when dealing with meshes that have multiple disconnected components or defects in the triangulation. Our framework also assumes that a template mesh of the shape is known for pose transfer. We plan to extend our framework for transferring poses between arbitrary shapes in future work.

**Societal Impacts.** Our generative model for poses and the pose transfer technique could potentially be misused for deepfakes. Developing robust guidelines and techniques to prevent such misuse is an important area for future research.

## Acknowledgments

We appreciate Minh Hieu Nguyen and Jisung Hwang for their support on creating figures. S. Yoo acknowledges the support of the Graduate School National Presidential Science Scholarship provided by Korea Student Aid Foundation. This work was supported by the NRF grant (RS-2023-00209723), IITP grants (RS-2022-II220594, RS-2023-00227592, RS-2024-00399817), and KEIT grant (RS-2024-00423625), all funded by the Korean government (MSIT and MOTIE), as well as grants from the DRB-KAIST SketchTheFuture Research Center, NAVER-Intel Co-Lab, Hyundai NGV, KT, and Samsung Electronics.

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

# Appendix

In the following appendix, we provide implementation details of our method, including dataset processing, network architectures, training details, and hyperparameter selection in Sec. A. We also present multi-view renderings of a pose transfer example showcased in Sec. 4.3 in Sec. B, the full list of evaluation metrics reported in Tab. 2 in Sec. C, and additional qualitative results from our pose transfer and generation experiments in Sec. D.

## A    Implementation Details

**Data.**    We use nine animal shapes from the DeformingThings4D-Animals [23] dataset in our experiments, specifically: BEAR, BUNNY, CANINE, DEER, DOG, ELK, FOX, MOOSE, and PUMA. We employ the first frame of the first animation sequence (alphabetically ordered) as the template for each animal, and the last frame of randomly sampled animation sequences for variations. Additionally, we use T-posed humanoid shapes from both the SMPL [28] and Mixamo [1] datasets as template meshes.

**Networks.**    We utilize Point Transformer layers from Zhao et al.[56] and Tang et al.[44] for implementing the pose extractor $g$ and the pose applier $h$. The network architectures for our cascaded diffusion models, as detailed in Sec. 4.6, are adapted from Koo et al. [21]. These models operate over $T = 1000$ timesteps with a linear noise schedule ranging from $\beta_1 = 1 \times 10^{-4}$ to $\beta_T = 5 \times 10^{-2}$. For model training, we employ the ADAM optimizer at a learning rate of $1 \times 10^{-3}$ and standard parameters. Our experiments are conducted on RTX 3090 GPUs (24 GB VRAM) and A6000 GPUs (48 GB VRAM).

For per-identity refinement modules, we set $\lambda_{\text{lap}} = 1.0$, $\lambda_{\text{edge}} = 1.0$, and $\lambda_{\text{reg}} = 5 \times 10^{-2}$ during training.

**Loss Functions.**    We provide the definitions of the loss functions used to supervise the training of our per-identity refinement module.

The Laplacian loss [27] is defined between two sets of vertices with the vertex-wise correspondences:

$$\mathcal{L}_{\text{lap}}\left(\mathbf{V}, \overline{\mathbf{V}}^T\right) = \mathbf{L}_T\left(\mathbf{V} - \overline{\mathbf{V}}^T\right), \tag{11}$$

where $\mathbf{V}$ is a set of new vertex coordinates, $\overline{\mathbf{V}}^T$ is a set of vertex coordinates of the target template mesh $\overline{\mathcal{M}}^T$, and $\mathbf{L}_T$ is the cotangent Laplacian of $\overline{\mathcal{M}}^T$.

Likewise, the edge length preservation loss [24] is defined as:

$$\mathcal{L}_{\text{edge}} = \sum_{\{i,j\}\in\mathcal{E}} |\|\mathbf{V}_i - \mathbf{V}_j\|_2 - \|\overline{\mathbf{V}}_i^T - \overline{\mathbf{V}}_j^T\|_2|, \tag{12}$$

where $\mathcal{E}$ is a set of edges comprising the target shapes and $\mathbf{V}_i$ and $\overline{\mathbf{V}}_i^T$ are the $i$-th vertex of a new set of vertex coordinates and that of $\overline{\mathcal{M}}^T$.

## B    Multi-View Renderings of Qualitative Results

While Fig. 5 showcases only single-view images, our method produces high-fidelity 3D shapes after pose transfer as shown in Fig. A10.

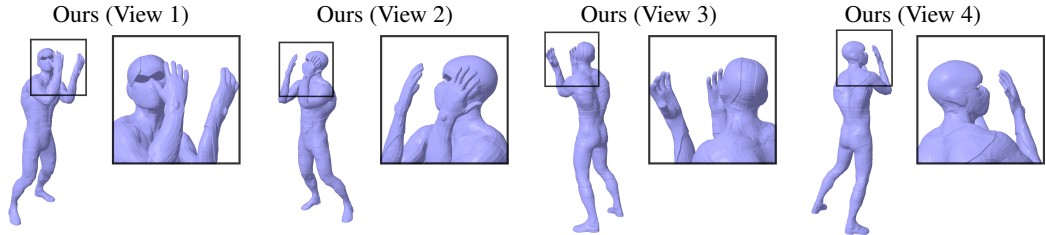

Ours (View 1)     Ours (View 2)     Ours (View 3)     Ours (View 4)

Figure A10: A pose transfer example showcased in Fig. 5, rendered from 4 different viewpoints.

## C   Full List of Quantitative Metrics Reported in Tab. 2

We report the full lists of evaluation metrics reported in Tab. 2 from the following page.

Table A4: Full list of FIDs reported in Tab. 2 (left)

| | Bear | Bunny | Canie | Deer | Dog | Elk | Fox | Moose | Puma |
|---|---|---|---|---|---|---|---|---|---|
| Vertices (Sec. 3.2) | | | | | | | | | |
| Bear | - | 3.47 | 11.53 | 6.76 | 6.14 | 8.40 | 4.92 | 5.26 | 9.80 |
| Bunny | 4.43 | - | 7.71 | 4.03 | 6.07 | 5.94 | 4.62 | 4.88 | 8.11 |
| Canie | 2.47 | 3.46 | - | 2.95 | 2.83 | 5.43 | 2.62 | 2.39 | 4.24 |
| Deer | 2.39 | 2.37 | 4.80 | - | 3.32 | 3.00 | 3.12 | 1.45 | 6.53 |
| Dog | 1.82 | 2.74 | 5.49 | 2.09 | - | 3.99 | 2.80 | 1.61 | 3.01 |
| Elk | 2.35 | 1.74 | 9.04 | 3.13 | 2.79 | - | 2.58 | 2.83 | 5.02 |
| Fox | 0.54 | 0.97 | 1.28 | 0.37 | 0.67 | 0.69 | - | 0.71 | 1.40 |
| Moose | 3.38 | 2.12 | 8.31 | 4.25 | 3.76 | 6.13 | 3.63 | - | 6.74 |
| Puma | 1.92 | 3.54 | 6.33 | 2.90 | 2.72 | 5.58 | 2.78 | 2.10 | - |
| Jacobians (Sec. 3.3) | | | | | | | | | |
| Bear | - | 6.77 | 1.16 | 0.61 | 0.46 | 0.27 | 0.73 | 1.20 | 0.42 |
| Bunny | 3.36 | - | 6.08 | 2.16 | 4.94 | 3.87 | 2.12 | 1.41 | 4.37 |
| Canie | 0.45 | 1.40 | - | 0.41 | 0.78 | 0.92 | 0.62 | 0.35 | 0.64 |
| Deer | 0.49 | 1.94 | 2.44 | - | 1.25 | 1.68 | 0.34 | 0.27 | 0.98 |
| Dog | 0.21 | 3.70 | 0.48 | 0.42 | - | 0.17 | 0.47 | 0.62 | 0.16 |
| Elk | 0.44 | 5.23 | 0.99 | 0.63 | 0.46 | - | 0.49 | 1.03 | 0.51 |
| Fox | 0.99 | 0.98 | 1.74 | 0.74 | 1.47 | 1.37 | - | 0.53 | 1.27 |
| Moose | 0.39 | 2.49 | 1.69 | 0.25 | 0.82 | 1.16 | 0.48 | - | 0.69 |
| Puma | 0.39 | 2.80 | 0.46 | 0.46 | 0.36 | 0.35 | 0.24 | 0.47 | - |
| Ours (Jacobians + Refinement (Sec. 3.4)) | | | | | | | | | |
| Bear | - | 3.83 | 1.10 | 0.60 | 0.56 | 0.30 | 0.58 | 0.92 | 0.41 |
| Bunny | 3.97 | - | 6.22 | 2.71 | 4.75 | 3.39 | 2.89 | 2.16 | 4.72 |
| Canie | 0.64 | 0.98 | - | 0.51 | 0.80 | 0.96 | 0.66 | 0.40 | 0.83 |
| Deer | 0.68 | 1.35 | 2.18 | - | 1.29 | 1.10 | 0.40 | 0.28 | 1.11 |
| Dog | 0.33 | 2.03 | 0.46 | 0.45 | - | 0.22 | 0.25 | 0.52 | 0.20 |
| Elk | 0.42 | 3.70 | 0.89 | 0.61 | 0.49 | - | 0.46 | 0.85 | 0.53 |
| Fox | 1.61 | 0.60 | 1.73 | 1.34 | 1.46 | 1.04 | - | 1.08 | 1.51 |
| Moose | 0.58 | 1.58 | 1.74 | 0.34 | 0.93 | 0.95 | 0.48 | - | 0.79 |
| Puma | 0.50 | 1.87 | 0.55 | 0.53 | 0.46 | 0.40 | 0.29 | 0.47 | - |

Table A5: Full list of KIDs reported in Tab. 2 (left)

| | Bear | Bunny | Canie | Deer | Dog | Elk | Fox | Moose | Puma |
|---|---|---|---|---|---|---|---|---|---|
| | | | | Vertices (Sec. 3.2) | | | | | |
| Bear | - | 2.56 | 8.87 | 4.95 | 4.46 | 6.15 | 3.65 | 3.61 | 6.95 |
| Bunny | 2.06 | - | 3.87 | 2.69 | 3.20 | 2.76 | 2.42 | 2.50 | 3.79 |
| Canie | 1.55 | 2.38 | - | 1.97 | 1.82 | 3.79 | 1.68 | 1.46 | 2.82 |
| Deer | 1.05 | 1.01 | 2.07 | - | 1.53 | 1.24 | 1.68 | 0.62 | 3.93 |
| Dog | 1.30 | 1.70 | 3.99 | 1.49 | - | 2.92 | 2.08 | 1.11 | 2.00 |
| Elk | 1.37 | 0.65 | 6.57 | 2.22 | 1.60 | - | 1.07 | 1.91 | 3.02 |
| Fox | 0.26 | 0.25 | 0.39 | 0.16 | 0.25 | 0.27 | - | 0.39 | 0.64 |
| Moose | 1.63 | 0.83 | 5.29 | 2.91 | 1.67 | 3.33 | 1.39 | - | 2.93 |
| Puma | 1.03 | 2.10 | 4.38 | 1.65 | 1.72 | 3.85 | 1.86 | 1.14 | - |
| | | | | Jacobians (Sec. 3.3) | | | | | |
| Bear | - | 2.36 | 0.29 | 0.21 | 0.20 | 0.01 | 0.17 | 0.27 | 0.03 |
| Bunny | 1.18 | - | 3.00 | 1.02 | 2.26 | 1.70 | 0.85 | 0.43 | 1.97 |
| Canie | 0.17 | 0.67 | - | 0.10 | 0.25 | 0.25 | 0.14 | 0.11 | 0.19 |
| Deer | 0.10 | 0.11 | 0.99 | - | 0.70 | 0.93 | 0.08 | 0.03 | 0.41 |
| Dog | 0.08 | 1.71 | 0.11 | 0.15 | - | 0.05 | 0.27 | 0.18 | -0.00 |
| Elk | 0.02 | 1.97 | 0.09 | 0.15 | 0.05 | - | 0.06 | 0.17 | 0.05 |
| Fox | 0.62 | 0.27 | 0.99 | 0.74 | 1.04 | 0.84 | - | 0.32 | 0.79 |
| Moose | 0.22 | 1.18 | 0.72 | 0.13 | 0.45 | 0.46 | 0.09 | - | 0.20 |
| Puma | 0.15 | 0.77 | 0.08 | 0.29 | 0.15 | 0.15 | 0.03 | 0.22 | - |
| | | | | Ours (Jacobians + Refinement (Sec. 3.4)) | | | | | |
| Bear | - | 1.26 | 0.29 | 0.23 | 0.20 | 0.01 | 0.08 | 0.27 | 0.07 |
| Bunny | 1.42 | - | 2.92 | 1.02 | 2.10 | 1.34 | 0.85 | 0.80 | 1.97 |
| Canie | 0.17 | 0.44 | - | 0.14 | 0.25 | 0.25 | 0.09 | 0.11 | 0.27 |
| Deer | 0.19 | 0.11 | 0.94 | - | 0.66 | 0.46 | 0.08 | 0.03 | 0.41 |
| Dog | 0.08 | 0.93 | 0.11 | 0.17 | - | 0.05 | 0.08 | 0.18 | 0.03 |
| Elk | 0.02 | 1.34 | 0.09 | 0.17 | 0.05 | - | 0.04 | 0.17 | 0.09 |
| Fox | 0.92 | 0.27 | 0.90 | 0.74 | 0.90 | 0.44 | - | 0.65 | 0.79 |
| Moose | 0.22 | 0.68 | 0.72 | 0.19 | 0.45 | 0.46 | 0.04 | - | 0.29 |
| Puma | 0.22 | 0.77 | 0.16 | 0.29 | 0.21 | 0.17 | 0.03 | 0.25 | - |

Table A6: Full list of ResNet classification accuracies reported in Tab. 2 (left)

| | Bear | Bunny | Canie | Deer | Dog | Elk | Fox | Moose | Puma |
|---|---|---|---|---|---|---|---|---|---|
| | | | | Vertices (Sec. 3.2) | | | | | |
| Bear | - | 81.33 | 37.58 | 58.00 | 72.75 | 53.17 | 65.83 | 54.00 | 41.33 |
| Bunny | 57.25 | - | 26.83 | 51.25 | 41.33 | 35.58 | 52.83 | 49.75 | 15.33 |
| Canie | 77.67 | 69.92 | - | 74.25 | 78.75 | 63.75 | 77.75 | 76.17 | 57.58 |
| Deer | 95.08 | 90.42 | 70.33 | - | 93.50 | 85.33 | 97.58 | 98.08 | 67.42 |
| Dog | 67.50 | 45.42 | 17.58 | 51.08 | - | 38.00 | 69.17 | 64.00 | 18.58 |
| Elk | 92.08 | 81.58 | 72.25 | 86.25 | 92.50 | - | 84.92 | 90.08 | 81.17 |
| Fox | 78.92 | 58.08 | 34.25 | 81.58 | 68.75 | 46.42 | - | 84.42 | 17.17 |
| Moose | 44.33 | 22.33 | 9.83 | 74.33 | 40.17 | 33.67 | 25.08 | - | 6.42 |
| Puma | 82.25 | 73.92 | 71.00 | 69.00 | 81.08 | 71.50 | 81.00 | 77.92 | - |
| | | | | Jacobians (Sec. 3.3) | | | | | |
| Bear | - | 93.50 | 85.50 | 92.08 | 95.58 | 95.50 | 96.42 | 93.83 | 91.50 |
| Bunny | 86.67 | - | 60.00 | 72.83 | 69.00 | 66.75 | 85.33 | 79.08 | 63.08 |
| Canie | 77.83 | 80.25 | - | 83.92 | 81.33 | 76.17 | 92.83 | 81.17 | 80.08 |
| Deer | 98.17 | 93.00 | 97.25 | - | 100.00 | 97.67 | 99.58 | 99.17 | 99.83 |
| Dog | 85.50 | 58.58 | 60.58 | 82.92 | - | 75.92 | 85.50 | 76.75 | 65.33 |
| Elk | 86.67 | 73.17 | 86.75 | 81.42 | 89.00 | - | 91.17 | 85.67 | 91.00 |
| Fox | 86.83 | 62.33 | 64.17 | 82.67 | 74.08 | 72.25 | - | 84.58 | 76.50 |
| Moose | 94.08 | 92.92 | 69.33 | 92.83 | 89.42 | 69.17 | 80.83 | - | 78.75 |
| Puma | 90.75 | 78.08 | 87.58 | 87.17 | 89.75 | 88.83 | 88.25 | 88.25 | - |
| | | | | Ours (Jacobians + Refinement (Sec. 3.4)) | | | | | |
| Bear | - | 91.67 | 92.50 | 93.00 | 96.58 | 95.75 | 97.42 | 94.42 | 94.67 |
| Bunny | 88.58 | - | 72.33 | 74.83 | 80.33 | 74.42 | 89.33 | 81.75 | 73.42 |
| Canie | 80.00 | 83.92 | - | 86.50 | 84.42 | 79.92 | 91.67 | 81.58 | 83.33 |
| Deer | 98.58 | 94.00 | 99.08 | - | 99.92 | 99.50 | 99.75 | 99.25 | 99.67 |
| Dog | 86.92 | 66.17 | 64.25 | 86.00 | - | 80.83 | 86.58 | 77.75 | 74.08 |
| Elk | 88.25 | 75.42 | 90.17 | 82.92 | 91.17 | - | 92.08 | 86.83 | 92.00 |
| Fox | 87.58 | 81.08 | 84.25 | 89.00 | 91.17 | 87.42 | - | 89.92 | 91.42 |
| Moose | 95.75 | 94.00 | 75.50 | 93.33 | 91.92 | 77.25 | 92.75 | - | 86.08 |
| Puma | 91.25 | 82.58 | 91.58 | 88.00 | 91.50 | 90.33 | 89.83 | 90.33 | - |

Table A7: Full list of FIDs reported in Tab. 2 (right)

| | Bear | Bunny | Canie | Deer | Dog | Elk | Fox | Moose | Puma |
|---|---|---|---|---|---|---|---|---|---|
| Vertices (Sec. 3.2) | | | | | | | | | |
| Bear | - | 6.48 | 30.77 | 12.79 | 12.01 | 20.99 | 10.43 | 24.06 | 12.48 |
| Bunny | 11.58 | - | 36.26 | 13.34 | 15.52 | 24.21 | 12.93 | 25.78 | 15.91 |
| Canie | 4.03 | 2.58 | - | 5.06 | 4.95 | 8.12 | 4.54 | 9.67 | 4.71 |
| Deer | 6.42 | 4.82 | 23.16 | - | 10.69 | 14.91 | 9.67 | 15.48 | 13.75 |
| Dog | 2.62 | 1.69 | 12.12 | 4.06 | - | 7.53 | 3.47 | 9.51 | 3.57 |
| Elk | 4.58 | 3.33 | 16.49 | 5.82 | 6.27 | - | 6.90 | 12.78 | 6.82 |
| Fox | 4.31 | 2.18 | 20.87 | 7.74 | 7.95 | 13.40 | - | 16.59 | 6.30 |
| Moose | 7.60 | 6.18 | 22.20 | 9.00 | 10.64 | 15.46 | 10.74 | - | 11.63 |
| Puma | 3.46 | 2.89 | 12.97 | 5.18 | 5.22 | 8.75 | 4.57 | 10.18 | - |
| Jacobians (Sec. 3.3) | | | | | | | | | |
| Bear | - | 2.33 | 9.66 | 2.12 | 4.62 | 14.47 | 1.82 | 0.80 | 2.74 |
| Bunny | 7.29 | - | 19.93 | 7.19 | 12.65 | 23.89 | 6.74 | 3.02 | 8.76 |
| Canie | 1.75 | 0.94 | - | 2.22 | 4.60 | 10.37 | 2.21 | 1.03 | 3.16 |
| Deer | 1.75 | 0.99 | 10.24 | - | 5.68 | 17.01 | 2.35 | 0.53 | 3.04 |
| Dog | 1.21 | 1.38 | 5.16 | 1.69 | - | 8.80 | 0.91 | 0.80 | 1.51 |
| Elk | 0.80 | 2.39 | 5.89 | 2.11 | 3.61 | - | 1.60 | 0.73 | 1.78 |
| Fox | 3.17 | 0.65 | 9.70 | 4.07 | 6.86 | 14.63 | - | 1.10 | 4.18 |
| Moose | 1.97 | 1.38 | 10.27 | 3.59 | 6.09 | 16.94 | 3.70 | - | 3.36 |
| Puma | 1.43 | 1.37 | 5.25 | 1.85 | 3.12 | 8.93 | 1.49 | 0.71 | - |
| Ours (Jacobians + Refinement (Sec. 3.4)) | | | | | | | | | |
| Bear | - | 1.46 | 9.01 | 2.20 | 4.49 | 13.71 | 2.14 | 0.85 | 2.72 |
| Bunny | 8.03 | - | 17.47 | 7.42 | 11.33 | 21.57 | 7.42 | 3.79 | 8.62 |
| Canie | 2.05 | 1.04 | - | 2.23 | 4.37 | 9.95 | 2.33 | 1.20 | 3.21 |
| Deer | 2.18 | 1.04 | 8.96 | - | 5.45 | 14.57 | 2.53 | 0.78 | 3.11 |
| Dog | 1.58 | 1.03 | 4.81 | 1.83 | - | 8.42 | 1.18 | 0.98 | 1.62 |
| Elk | 0.99 | 1.84 | 5.57 | 2.09 | 3.58 | - | 1.65 | 0.79 | 1.79 |
| Fox | 3.91 | 1.61 | 7.59 | 3.85 | 5.53 | 12.00 | - | 2.14 | 3.79 |
| Moose | 2.36 | 1.25 | 9.65 | 3.72 | 5.98 | 16.01 | 3.46 | - | 3.50 |
| Puma | 1.66 | 1.23 | 5.09 | 1.94 | 3.16 | 8.59 | 1.60 | 0.84 | - |

Table A8: Full list of KIDs reported in Tab. 2 (right)

| | Bear | Bunny | Canie | Deer | Dog | Elk | Fox | Moose | Puma |
|---|---|---|---|---|---|---|---|---|---|
| Vertices (Sec. 3.2) | | | | | | | | | |
| Bear | - | 3.53 | 15.17 | 7.02 | 5.90 | 10.50 | 5.51 | 10.86 | 6.91 |
| Bunny | 5.15 | - | 17.82 | 6.40 | 7.48 | 11.71 | 6.10 | 12.39 | 7.15 |
| Canie | 1.74 | 1.33 | - | 2.21 | 2.05 | 3.52 | 1.99 | 4.47 | 2.00 |
| Deer | 3.41 | 2.80 | 14.00 | - | 6.58 | 8.71 | 5.83 | 8.33 | 8.77 |
| Dog | 1.31 | 0.83 | 5.61 | 1.88 | - | 3.36 | 1.66 | 4.15 | 1.65 |
| Elk | 2.13 | 1.46 | 7.88 | 2.70 | 2.85 | - | 3.04 | 5.55 | 3.25 |
| Fox | 2.28 | 1.10 | 11.40 | 4.44 | 4.24 | 7.07 | - | 8.84 | 3.36 |
| Moose | 3.49 | 2.77 | 10.45 | 4.22 | 5.14 | 6.91 | 4.92 | - | 4.88 |
| Puma | 1.48 | 1.39 | 5.73 | 2.24 | 2.10 | 3.63 | 2.09 | 4.46 | - |
| Jacobians (Sec. 3.3) | | | | | | | | | |
| Bear | - | 0.66 | 5.20 | 1.35 | 2.61 | 7.83 | 0.95 | 0.23 | 1.37 |
| Bunny | 2.85 | - | 10.52 | 3.68 | 6.35 | 12.27 | 2.81 | 0.97 | 3.84 |
| Canie | 0.65 | 0.39 | - | 1.29 | 2.25 | 5.33 | 0.77 | 0.24 | 1.35 |
| Deer | 0.72 | 0.17 | 6.97 | - | 4.06 | 12.34 | 1.33 | 0.15 | 1.45 |
| Dog | 0.52 | 0.60 | 2.90 | 1.13 | - | 4.84 | 0.40 | 0.19 | 0.67 |
| Elk | 0.20 | 0.82 | 3.32 | 1.38 | 1.97 | - | 0.77 | 0.14 | 0.85 |
| Fox | 1.96 | 0.27 | 6.08 | 2.89 | 4.52 | 8.89 | - | 0.64 | 2.48 |
| Moose | 0.79 | 0.54 | 5.68 | 2.43 | 3.41 | 9.92 | 1.86 | - | 1.51 |
| Puma | 0.78 | 0.53 | 2.99 | 1.29 | 1.77 | 5.13 | 0.74 | 0.38 | - |
| Ours (Jacobians + Refinement (Sec. 3.4)) | | | | | | | | | |
| Bear | - | 0.35 | 4.87 | 1.39 | 2.57 | 7.38 | 1.16 | 0.28 | 1.38 |
| Bunny | 3.12 | - | 9.03 | 3.84 | 5.52 | 10.76 | 3.11 | 1.30 | 3.71 |
| Canie | 0.79 | 0.41 | - | 1.26 | 2.11 | 5.04 | 0.83 | 0.33 | 1.39 |
| Deer | 0.93 | 0.35 | 5.93 | - | 3.74 | 10.15 | 1.40 | 0.28 | 1.45 |
| Dog | 0.71 | 0.42 | 2.67 | 1.20 | - | 4.54 | 0.55 | 0.28 | 0.72 |
| Elk | 0.29 | 0.59 | 3.12 | 1.39 | 1.97 | - | 0.77 | 0.18 | 0.86 |
| Fox | 2.23 | 0.84 | 4.52 | 2.46 | 3.47 | 6.96 | - | 1.19 | 2.11 |
| Moose | 0.97 | 0.44 | 5.31 | 2.52 | 3.35 | 9.27 | 1.68 | - | 1.61 |
| Puma | 0.91 | 0.48 | 2.91 | 1.34 | 1.81 | 4.91 | 0.80 | 0.46 | - |

Table A9: Full list of ResNet classification accuracies reported in Tab. 2 (right)

| | Bear | Bunny | Canie | Deer | Dog | Elk | Fox | Moose | Puma |
|---|---|---|---|---|---|---|---|---|---|
| Vertices (Sec. 3.2) | | | | | | | | | |
| Bear | - | 59.25 | 8.17 | 24.08 | 48.75 | 15.50 | 36.00 | 20.75 | 30.17 |
| Bunny | 38.50 | - | 0.83 | 15.00 | 21.25 | 9.75 | 24.33 | 13.83 | 5.17 |
| Canie | 72.58 | 78.42 | - | 56.67 | 71.92 | 39.50 | 61.50 | 35.67 | 66.58 |
| Deer | 92.42 | 84.83 | 65.58 | - | 88.25 | 75.58 | 82.00 | 85.00 | 73.83 |
| Dog | 66.75 | 45.00 | 6.17 | 28.08 | - | 13.92 | 51.17 | 24.25 | 14.75 |
| dragonOLO | 99.67 | 97.92 | 82.25 | 100.00 | 99.58 | - | 99.67 | 94.83 | 96.33 |
| Elk | 86.83 | 71.33 | 32.50 | 62.92 | 74.50 | 57.50 | - | 44.50 | 81.92 |
| Fox | 64.42 | 62.83 | 10.58 | 47.17 | 54.50 | 22.42 | 66.67 | - | 13.25 |
| Moose | 32.58 | 15.42 | 1.33 | 41.50 | 19.83 | 11.50 | 13.00 | 35.83 | - |
| Jacobians (Sec. 3.3) | | | | | | | | | |
| Bear | - | 91.58 | 77.75 | 90.33 | 93.92 | 82.67 | 94.92 | 94.83 | 90.67 |
| Bunny | 90.92 | - | 42.67 | 70.17 | 74.33 | 44.92 | 84.17 | 84.08 | 68.25 |
| Canie | 85.17 | 87.42 | - | 77.83 | 84.67 | 68.08 | 96.00 | 85.75 | 89.58 |
| Deer | 99.67 | 96.75 | 99.58 | - | 99.92 | 99.58 | 100.00 | 99.75 | 100.00 |
| Dog | 90.92 | 69.75 | 69.00 | 83.08 | - | 78.92 | 93.17 | 87.83 | 81.33 |
| Elk | 94.50 | 80.67 | 94.42 | 78.58 | 92.42 | - | 95.00 | 91.92 | 92.50 |
| Fox | 89.17 | 76.33 | 59.92 | 79.25 | 78.67 | 39.08 | - | 92.17 | 89.58 |
| Moose | 97.00 | 91.58 | 57.92 | 84.50 | 86.83 | 45.17 | 68.67 | - | 85.33 |
| Puma | 93.67 | 84.42 | 93.42 | 84.75 | 92.42 | 95.25 | 89.83 | 90.75 | - |
| Ours (Jacobians + Refinement (Sec. 3.4)) | | | | | | | | | |
| Bear | - | 90.83 | 84.75 | 90.58 | 96.08 | 86.75 | 96.42 | 94.75 | 95.92 |
| Bunny | 91.08 | - | 56.00 | 72.33 | 81.67 | 53.17 | 88.17 | 86.00 | 77.42 |
| Canie | 87.75 | 89.08 | - | 79.33 | 85.92 | 69.08 | 95.25 | 86.50 | 90.75 |
| Deer | 99.83 | 98.00 | 99.92 | - | 100.00 | 100.00 | 100.00 | 99.67 | 99.83 |
| Dog | 91.83 | 73.50 | 76.92 | 84.50 | - | 85.67 | 93.58 | 89.17 | 87.08 |
| Elk | 95.00 | 81.92 | 95.00 | 81.75 | 92.50 | - | 95.00 | 93.17 | 93.75 |
| Fox | 88.17 | 85.58 | 78.58 | 82.92 | 91.25 | 61.33 | - | 90.50 | 94.75 |
| Moose | 97.08 | 92.00 | 67.08 | 84.92 | 91.08 | 55.83 | 86.92 | - | 89.75 |
| Puma | 93.75 | 87.08 | 94.50 | 86.75 | 92.08 | 95.50 | 91.17 | 92.75 | - |

## D  Additional Qualitative Results

We showcase additional qualitative results from the pose transfer and pose generation experiments below.

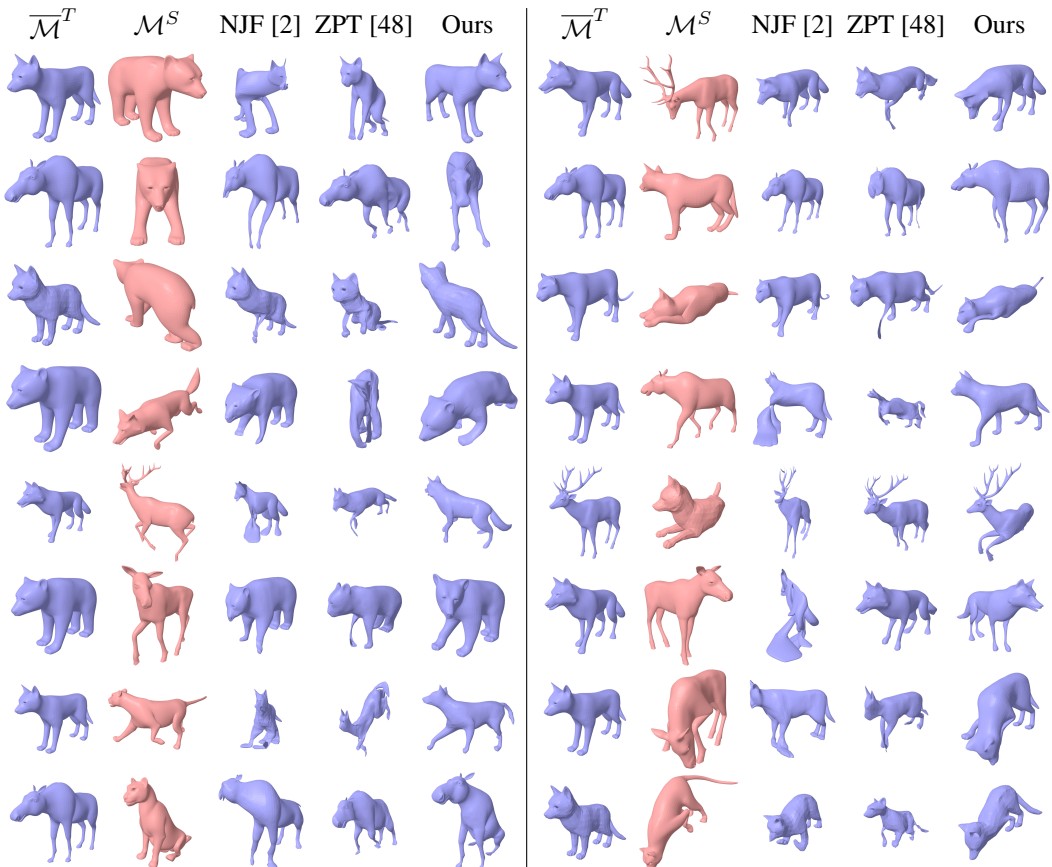

Figure A11: Qualitative results of pose transfer across DeformingThings4D animals [23].

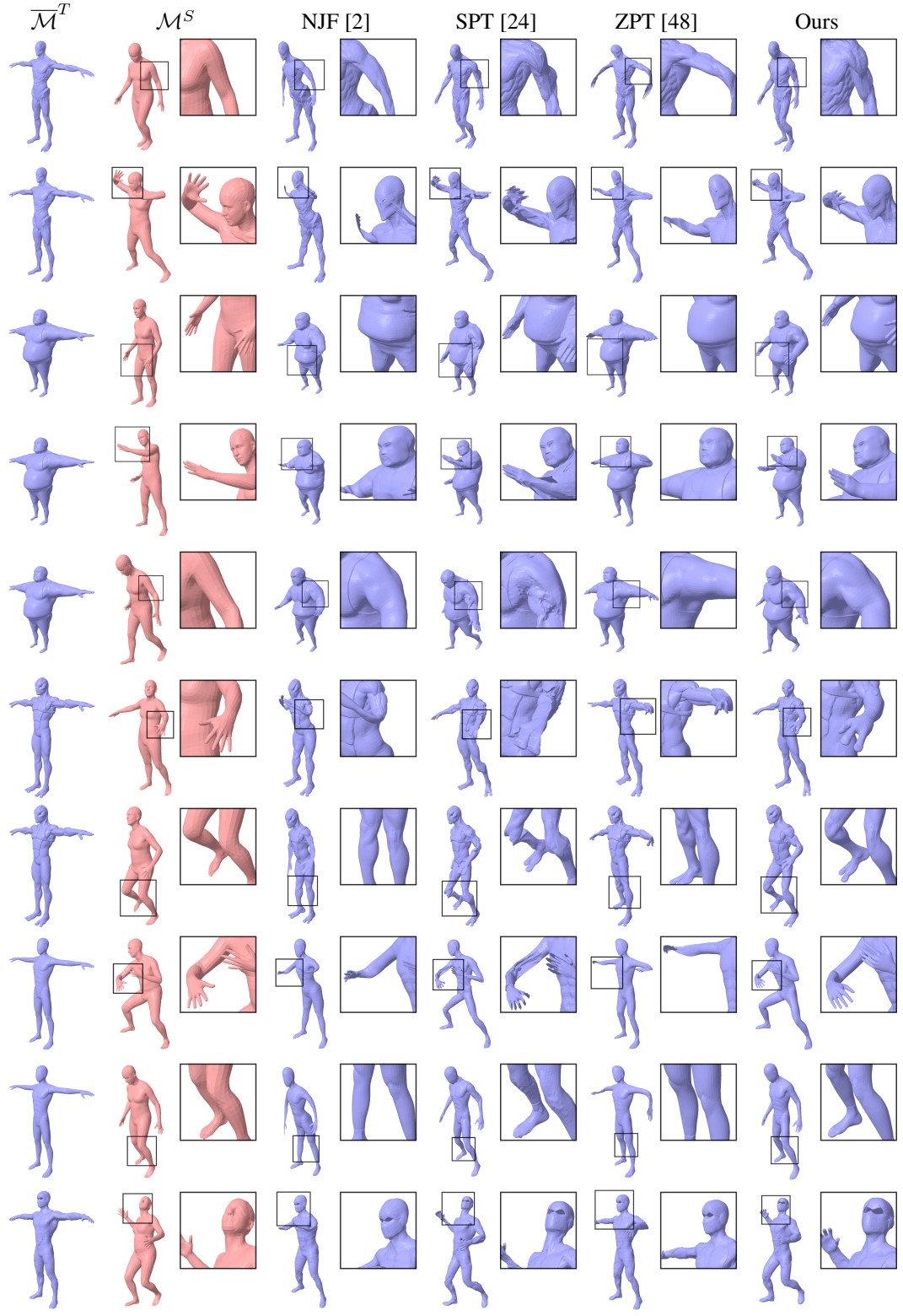

Figure A12: Qualitative results of pose transfer from a SMPL [28] mesh to Mixamo characters [1]. Best viewed when zoomed in.

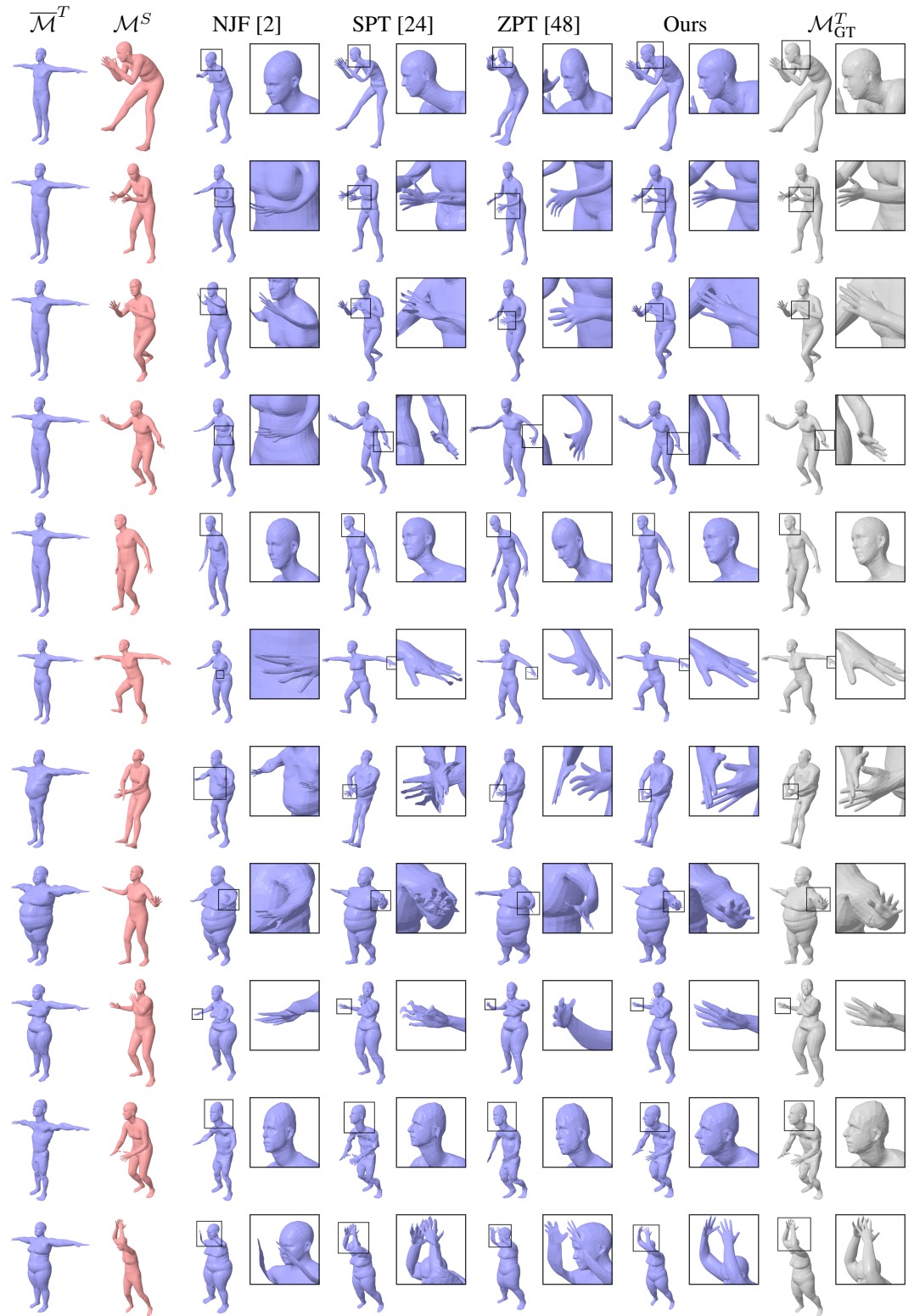

Figure A13: Qualitative results of pose transfer across different SMPL [28] human body shapes. Best viewed when zoomed-in.

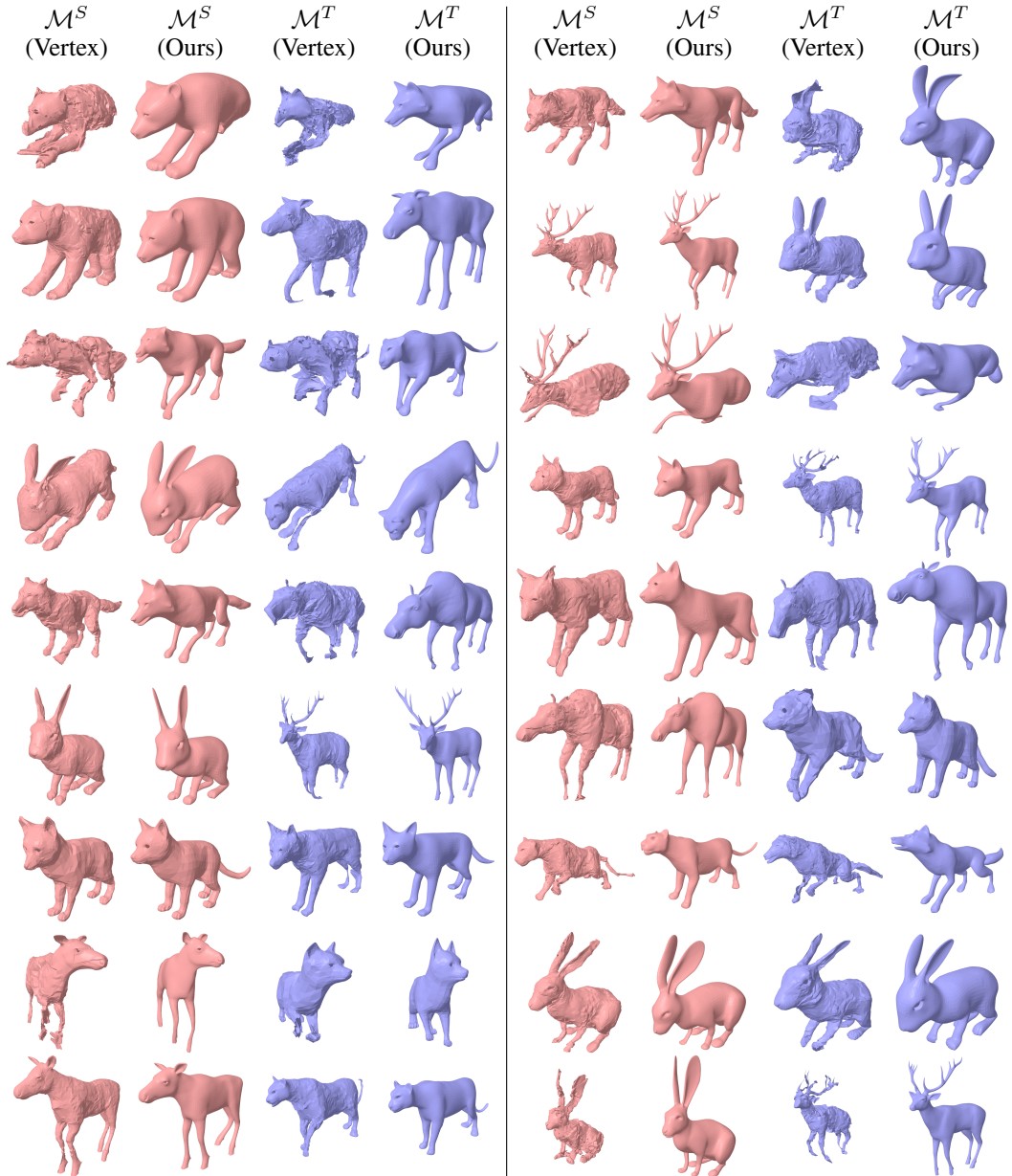

Figure A14: Unconditional generation results. Each row illustrates the outcome of directly applying the generated poses to the source shape $\mathcal{M}^S$ and then transferring them to various target shapes $\mathcal{M}^T$.

