# OpenReview forum: "Neural Pose Representation Learning for Generating and Transferring Non-Rigid Object Poses"
_NeurIPS.cc/2024/Conference — NeurIPS 2024 poster_

### Official Review · Reviewer_RKb2 · 2024-07-07

**Soundness:** 3
**Presentation:** 2
**Contribution:** 3
**Rating:** 5
**Confidence:** 4

**Summary:**

The authors propose a skeleton-free pipeline for implicit 3D pose representation and transfer. They introduce the prediction of Jacobian fields to achieve shape-preserving representation, which facilitates the application of transferred poses. To enhance accuracy, a per-identity refinement step is included, utilizing intrinsic-preserving loss. Additionally, a cascaded diffusion model is employed on the compact pose representation to generate novel poses.

**Strengths:**

1.The use of Jacobian fields for shape representation is innovative.
2.The method of incorporating an intermediate step for intrinsic preservation is justified.
3.Training a diffusion model on extracted implicit poses further demonstrates the effectiveness of the pose representation.

**Weaknesses:**

1.FPS is utilized to sample keypoints as a basis for pose representation. However, for complex poses, FPS-sampled keypoints may not be evenly distributed on the surface due to a lack of awareness of mesh connectivity, potentially resulting in poor pose representation.
2.Given that the pose is represented by keypoints' coordinates and corresponding latent features, the shape information is inherently entangled in the keypoints' coordinates. This could lead to the shape information being encoded in the latent features.

**Questions:**

1.The proposed method may face limitations when animating non-T-pose meshes if it relies heavily on T-pose assumptions for pose estimation and transfer.
2.It would be beneficial to demonstrate pose transfer results for both source and target meshes from the Mixamo dataset and compare these with results from the ECCV 2022 paper "Skeleton-free Pose Transfer for Stylized 3D Characters" to assess performance and efficacy.
3.Comparing the training and inference times with baseline methods would provide insights into the computational efficiency of the proposed approach.

**Limitations:**

Yes, they are discussed.

---

> ### Author Rebuttal · Authors · 2024-08-06
>
> Dear reviewer RKb2,
>
> Thank you for your positive comments. We are especially grateful for your recognition of the use of Jacobian fields as "innovative" and for acknowledging that our approach is well "justified". We provide our responses to your queries below.
>
> **Impact of the Sampling Method and Number of Keypoints.**
> We appreciate your suggestion for an in-depth analysis. We used FPS for keypoint extraction due to its simplicity and efficacy, which have made it a popular choice in geometry analysis (e.g., PointNet++ [Qi et al., NeurIPS 2017], KeypointDeformer [Jakab et al., CVPR 2021], DeepMetaHandles [Liu et al., CVPR 2021]). Here, we investigate the impact of the number of keypoints and their sparsity. We trained our model using SMPL and DeformingThings-Animals meshes, adjusting the pose extractor to extract 50, 25, and 10 keypoints, respectively. We evaluated the Point-wise Mesh Euclidean Distance (PMD) for SMPL meshes and Fréchet inception distance (FID) for DeformingThings4D-Animals meshes, following our paper. The results are summarized in Tables 1 and 2. Notably, the model trained with just 10 keypoints still outperforms the pretrained SPT model in the pose transfer using SMPL meshes. Please refer to Figures 3 and 4 in the PDF file. We will include this analysis in the future revision.
>
> **Table 1: PMD measured on SMPL pose transfer experiments with varying number of keypoints. "Ours-$N$" denotes a variant of our network trained to extract $N$ keypoints.**
>
> | Method      | SPT | Ours-10 | Ours-25 | Ours-50 | **Ours-100** |
> |------------- |-----|---------|---------|---------|--------------|
> | PMD ($\times 10^{-3}$)| 0.28 | 0.20    | 0.17    | 0.17    | **0.13**     |
>
> **Table 2: FID measured on meshes whose poses are transferred from class **bear3EP** of the **DeformThings** dataset. "Ours-$N$" denotes a variant of our network trained to extract $N$ keypoints.**
>
> | Method   | Ours-10 | Ours-25 | Ours-50 | **Ours-100** |
> |----------|---------|---------|---------|--------------|
> | FID ($\times10^{-2}$) | 1.25    | 0.87    | 0.83    | **0.72**     |
>
> **Potential Drawback of Representing Poses as Keypoints.** To prevent the leakage of source shape details into target shapes, we use Jacobian fields instead of vertex coordinates when extracting our pose representation. This way, our network predicts per-triangle *local transformations* to deform the given template to match the pose example. Furthermore, training the refinement module by optimizing intrinsic-preservation losses helps preserve intricate local details. Please refer to Figure 5 in the attached PDF file for qualitative results of the ablation study. We will include more results in the revised version.
>
> **Assumption on T-Poses.** Thank you for pointing this out. It is true that our method requires a canonical pose for an identity (e.g., T-pose for humanoids). However, we believe our method remains practical for its intended purpose since default or canonical poses of deformable objects have been widely used not only for humanoids but also for animal shapes, including in works like SMAL [Zuffi et al., CVPR 2017], A-CSM [Kulkarni et al., CVPR 2020], BARC [Ruegg et al., CVPR 2022], BITE [Ruegg et al., CVPR 2023], VAREN [Zuffi et al., CVPR 2024], and 3D Fauna [Li et al., CVPR 2024]. These studies define a canonical pose as a shape standing still with its legs straight for quadrupeds. Nonetheless, there may be cases where obtaining, or even defining a canonical pose of an object is challenging. As such, lifting the necessity of template shapes is one of the directions that we are heading to.
>
> **Pose Transfer between Mixamo Meshes.** We appreciate your suggestion. We downloaded 28 motions in the test split of SPT [Liao et al., ECCV 2022] from the Mixamo repository, comprising a total of 3,025 frames. We followed the preprocessing steps of SPT to extract meshes from skeleton configurations. Since the pretrained SPT model was trained on a larger dataset consisting of shapes from AMASS, Mixamo, and RigNet, we re-trained the model to analyze its performance in the same problem setup as ours. For training, we used 3,025 different poses of a single character. After training, we used 300 poses transferred to 8 different characters to compute Point-wise Mesh Euclidean Distance (PMD) by leveraging the ground-truth correspondences. We report the results in Table 3. Please refer to Figure 6 in the attached PDF file for qualitative results. As illustrated, our method better retains the smoothness and details of the surface even when transferring poses involving articulations of limbs. This is also reflected in lower PMD in Table 3. We will add these results in the revision.
>
> **Table 3: PMD measured on Mixamo pose transfer experiments.**
> | Method       | SPT  | **Ours** |
> |--------------|------|----------|
> | PMD ($\times10^{-3}$) | 3.42 | **2.28** |
>
> **Training \& Inference Time Comparisons.** We summarize the training and inference time in Table 4. The inference time includes the time to transfer a pose of one identity to another via network forward passes. We used Mixamo meshes with approximately 10K vertices and 20K faces for evaluation. Note that for SPT, we measured the inference time using the official pretrained model. Our method demonstrates better runtime performance than SPT while outperforming NJF and ZPT in terms of pose transfer accuracy. We will include this analysis in the revised version of our paper.
>
> **Table 4: Training and inference time comparison**
> | Method            | NJF  | SPT  | ZPT  | **Ours** |
> |-------------------|------|------|------|----------|
> | Training          | 2h   | 20h  | 9h   | 8h       |
> | Inference (per pair) | 0.005s | 1s   | 0.004s | 0.03s    |

---

> ### Comment · Reviewer_RKb2 · 2024-08-12
> **Response to rebuttal**
>
> Given the additional clarification and experimental results, I am more inclined to recommend acceptance of this paper. However, I am also want to see the potential drawbacks of using jacobian fields for shape representation and failure cases caused by simply using FPS as basis(for extreme poses and loose clothes). Also, the performance seems drop a lot for SPT after your retraining, is it possible to train on the same dataset to compare with their reported results?

---

> > ### Author Response · Authors · 2024-08-13
> >
> > Dear reviewer RKb2,
> >
> > We sincerely appreciate your positive feedback. Due to space constraints, we were unable to fully address all of your comments in the rebuttal. Please allow us to further address them below.
> >
> > **Potential Drawbacks of Using Jacobian Fields**
> >
> > We did not encounter noticeable failure cases associated with the use of Jacobian Fields. Our ablation study demonstrated that the overall quality of pose transfer improves with Jacobian Fields, both quantitatively and qualitatively. We will include additional qualitative results from the ablation study in the final revision.
> >
> > One potential drawback of using Jacobian Fields is the requirement for differential operators, such as the cotangent Laplacian and gradient operator. These differential operators may not be available for meshes with multiple connected components, as noted in the original NJF paper [Aigerman et al., ACM ToG 2022]. We will also clarify this limitation in the revision.
> >
> >
> > **Failure Cases due to FPS**
> >
> > We also did not observe specific failure cases related to the use of FPS keypoint sampling. While we believe that simple FPS point sampling is sufficient for our framework, in the final revision, we will further test our method with other keypoint sampling techniques, such as uniform sampling, and report the results. Thank you for your detailed comments and suggestions. Please understand that the new experiment requires more time than is available in the remaining discussion phase.
> >
> >
> > **Comparison with SPT**
> >
> > We would like to clarify that the results reported above are based on training both SPT and our model with shapes of a **single** identity, which aligns with our problem setup. In our work, we focused on learning pose representations from shapes of a single identity and transferring them to a new identity. In contrast, SPT uses a much larger dataset that includes a variety of identities as training data, which is why we could not directly apply their training/test splits to our method. While it is technically feasible to extend our framework to train with multiple identities, we found that it would require substantial changes to our algorithm. Also, please note that our method requires preprocessing of meshes to compute differential operators, and some Mixamo shapes fail during this preprocessing. In the revision, we promise to extend our method to train the networks with multiple identities and report the results using the closest train/test splits to those of SPT (excluding models that fail in the preprocessing).
> >
> > We would also like to emphasize that, while this is not a perfect apple-to-apple comparison due to the different train/test splits, the pose transfer accuracy of our method trained with a **single** identity ($2.28 \times 10^{-3}$ PMD) is comparable to that of SPT, which was trained with a much larger dataset ($2.39 \times 10^{-3}$ PMD, as shown in Table 4 of the SPT paper). This demonstrates the effectiveness of our method, even with a significantly smaller training dataset.

---

### Official Review · Reviewer_xBSu · 2024-07-11

**Soundness:** 4
**Presentation:** 4
**Contribution:** 4
**Rating:** 9
**Confidence:** 5

**Summary:**

This paper presents a novel representation learning framework for pose estimation which is disentangled from the identity of the object. This implicit representation is used for generating and transferring poses using a cascade diffusion model. A keypoint-based hybrid pose representation with a sparse mesh is used to ensure it is compact enough for the generative model. Moreover, the authors use Jacobian fields as the shape representation, allowing their implicit deformation network to take mesh face coordinates as input as opposed to vertex coordinates. The latent representations learned from this are used in a self-supervised per-identity refinement module for further improvement. This allows the proposed network to transfer motion sequences from one object to another. Moreover, these representations can be used to animate previously unseen characters. The authors provide extensive experiments with both quantitative and qualitative ablations, showing the superiority of their approach over three baseline networks, using both animals and humanoid objects from different datasets.

**Strengths:**

- The paper is very well-written with clear descriptions of motivation, research question, methodology, mathematical formulations, and experiments.
- There are extensive experimental results to establish the superiority of the proposed method. In particular, the qualitative results clearly help visualize the improvements.
- It is very easy to read

**Weaknesses:**

- No weaknesses

**Questions:**

-The results look very promising, and  I am curious how this proposed method can be transferred to images or videos (with textures and backgrounds) insead of only mesh generation. Do you have any insights about how this will work and how your method can be adapted for this scenario?

**Limitations:**

The limitations of Jacobian fields are discussed in the conclusion

---

> ### Author Rebuttal · Authors · 2024-08-06
>
> Dear reviewer xBSu,
>
> We highly appreciate your positive comments on our work, particularly regarding the writing and experiment procedures. Our response to your question can be found below.
>
> **Extension to Images and Videos.** We strongly agree that our work can be extended to other modalities, such as images and videos. For instance, the proposed pose representation and our cascaded diffusion model can be utilized as a prior for 3D-aware image editing tasks where estimating motion trajectories and handling occlusions are crucial. Specifically, we believe our method can be adapted to recent drag-based image editing techniques, such as DragGAN [Pan et al., SIGGRAPH 2023] and DragDiffusion [Shi et al., CVPR 2024]. When a user inputs a set of drag-based instructions describing complex articulations of animals that cannot be resolved in 2D, we may consider lifting the object to 3D by using a single-view 3D reconstruction model (e.g., Zero-1-to-3 [Liu et al., ICCV 2023]). The coarse 3D delegate of the image can then be given to our model as an input, which extracts a pose representation from the shape. As the shape is edited following the instructions, our diffusion model, trained on a collection of pose representations, can provide guidance signals via Score Distillation Sampling [Poole et al., ICLR 2023] to ensure the resulting deformation remains realistic while faithfully reflecting the user's input. Nonetheless, this is one possible scenario where our method can be applied, and we hope to extend it to a variety of applications in the future.

---

> > ### Comment · Reviewer_xBSu · 2024-08-12
> >
> > Thank you for your response. After reading other reviewers' comments and looking at the additional experiments the authors have provided, I believe this work shows significant progress in an important field for the research community. The extensive qualitative results clearly show the advantages of the proposed approach. The authors' responses to other reviews are also satisfactory, and I stand by my original score for this paper. I strongly recommend it for acceptance.

---

> > > ### Author Response · Authors · 2024-08-13
> > >
> > > Dear reviewer xBSu,
> > >
> > > We highly appreciate your efforts in reviewing our work. Your valuable feedback has inspired us to explore future research directions by extending our work to other modalities, including images and videos.

---

### Official Review · Reviewer_skND · 2024-07-13

**Soundness:** 3
**Presentation:** 3
**Contribution:** 3
**Rating:** 6
**Confidence:** 4

**Summary:**

This paper introduces a novel 3D generative model for pose-identity disentangled representation of 3D shapes. It proposes to use a set of keypoints with features to represent the pose of a 3D shape and learn a pose-extractor and pose-applier to accomplish pose transfer between instances. Experiments are conducted on widely used benchmarks with two categories: animals and humans, showing superiority of the proposed method compared to the baselines.

**Strengths:**

- The studies task is novel and important, with broad applications in different communities.
- The proposed method is sound and effective. Using a dense set of keypoints to represent the pose of 3D shape is natural and well-motivated, which also results good results in the experiments.
- The evaluation is extensive and informative. Experiments are conducted on widely-used datasets, with comparisons against enough baselines. For baselines without the code, the paper provides results with own implementation. Ablation studies show the effectiveness of the proposed components.
-  The writing of the paper is mostly clear, illustrating their method and evaluation in a clean manner.

**Weaknesses:**

- Although the exposition is mostly good to me, there still remains some questions that are not fully addressed. See the questions below.
- No video results or animated results are provided. Therefore it's hard to evaluate the 3D/4D generated results (e.g., whether the result is 3D consistent from any view or whether the temporal consistency is good).
- No qualitative results for the ablation study is provided.
- It would be better if analysis on the keypoints can be provided. This might include analysis on the number of keypoints and the way to sample it.

**Questions:**

- How would the number of keypoints affect the result? What would be the fewest number of keypoints that is still feasible for this pipeline?
- Regarding the experiments:
  - What is the detailed setting of NJF? If I understand correctly, NJF takes the pose parameter of the target pose as input. What should be the pose parameter for the deformingthings4d dataset?
  - How is the texture of the generated mesh predicted? In L265, the authors mentioned to use FID to evaluate the visual fidelity, yet how is this implemented, given that the generated mesh seems to not have texture? Also, how is the camera determined to generate the images used to calculate FID?
- How is the "template" mesh defined in the framework, especially for categories without a template, such as animals? For humans, I understand that you may use T-pose human mesh as the template, but what would the case for the animals?

**Limitations:**

No failure cases are discussed in the paper. Under what condition would the proposed method fail?

---

> ### Author Rebuttal · Authors · 2024-08-06
>
> Dear reviewer skND,
>
> Thank you for recognizing our work as a study of a "novel" and "important" task, and for appreciating our "sound" and "well-motivated" approach. We have carefully reviewed your queries and provide our responses below.
>
> **Temporal Consistency.**
> Thank you for providing the constructive comment. In our work, we mainly focus on learning a transferable pose representation from pose variations of a *single* identity, which is more challenging than our prior works. While our framework has not been explicitly designed for motion transfer--such as by using a temporal attention mechanism or training with motion supervision--our framework can transfer motions by transferring each frame individually as demonstrated in Figure 1 in the PDF file in our global response. We applied the One Euro Filter, a simple and easy-to-implement filter, to refine the smoothness of motions.
> We plan to extend our work for motion transfer in the future. We will include more results in the revised version.
>
> **Spatial Consistency.** While we only showed single-view images in our paper, our method produces meshes consistent in 3D as illustrated in Figure 2 in the accompanied PDF file. In particular, we rendered one of our results shown in Figure 5 of the main paper from 4 different viewpoints. We will add more images rendered from various viewpoints in the revision.
>
> **Qualitative Results for Ablation Study.** The qualitative result can be found in Figure 5 in the PDF file. As shown, directly using vertices results in low-quality meshes with noticeable distortions and artifacts. Our base model employing Jacobian fields produces high-quality shapes, which can be further improved by using the proposed refinement module. Note the preservation of intricate details near limbs. We will add the analysis and more results in the upcoming revision.
>
> **Impact of the Sampling Method and Number of Keypoints.**
> We appreciate your suggestion for an in-depth analysis. We used FPS for keypoint extraction due to its simplicity and efficacy, which have made it a popular choice in geometry analysis (e.g., PointNet++ [Qi et al., NeurIPS 2017], KeypointDeformer [Jakab et al., CVPR 2021], DeepMetaHandles [Liu et al., CVPR 2021]). Here, we investigate the impact of the number of keypoints and their sparsity. We trained our model using SMPL and DeformingThings-Animals meshes, adjusting the pose extractor to extract 50, 25, and 10 keypoints, respectively. We evaluated the Point-wise Mesh Euclidean Distance (PMD) for SMPL meshes and Fréchet inception distance (FID) for DeformingThings4D-Animals meshes, following our paper. The results are summarized in Tables 1 and 2. Notably, the model trained with just 10 keypoints still outperforms the pretrained SPT model in the pose transfer using SMPL meshes. Please refer to Figures 3 and 4 in the PDF file. We will include this analysis in the future revision.
>
> **Table 1: PMD measured on SMPL pose transfer experiments with varying number of keypoints. "Ours-$N$" denotes a variant of our network trained to extract $N$ keypoints.**
>
> | Method      | SPT | Ours-10 | Ours-25 | Ours-50 | **Ours-100** |
> |------------- |-----|---------|---------|---------|--------------|
> | PMD ($\times 10^{-3}$)| 0.28 | 0.20    | 0.17    | 0.17    | **0.13**     |
>
> **Table 2: FID measured on meshes whose poses are transferred from class **bear3EP** of the **DeformThings** dataset. "Ours-$N$" denotes a variant of our network trained to extract $N$ keypoints.**
>
> | Method   | Ours-10 | Ours-25 | Ours-50 | **Ours-100** |
> |----------|---------|---------|---------|--------------|
> | FID ($\times10^{-2}$) | 1.25    | 0.87    | 0.83    | **0.72**     |
>
> **Clarification on NJF.** In our experiments, we followed the setup of the morphing humans experiment in [Aigerman et al., ACM ToG 2022] where an MLP takes PointNet latents encoded from both source and target shapes and predicts a Jacobian field of the source shape that morphs it to the target shape. We did not use the setup of the re-posing humans experiment as our work does not rely on parameterizations, such as skeletons. We will clarify this in the revision.
>
> **Clarification on FID Computation.** Unlike 2D images and videos with abundant reference data collected over the Internet, assessing the realism of 3D shapes remains challenging as there is no standardized way. We followed previous work including MeshDiffusion [Liu et al., ICLR 2023], 3DShape2VecSet [Zhang et al., ACM ToG 2023], MeshGPT [Siddiqui et al., CVPR 2024], and Make-A-Shape [Hui et al., ICML 2024] that compute FID using images rendered from multiple viewpoints without texture. Specifically, we rendered images from 4 viewpoints (front, back, left, and right) at zero elevation.
> Following MeshDiffusion [Liu et al., ICLR 2023], we applied a grayscale diffuse material for reference and output shapes and applied Phong shading to obtain images with depth cues. We will include these details of the evaluation settings in the revised version.
>
> **Template Meshes for Animals.** For quadrupeds, used in our experiments and comprising a large portion of animal species, we consider shapes standing with straight legs as the animal equivalent of the T-pose for humanoids. This convention has been used in several works, including SMAL [Zuffi et al., CVPR 2017], A-CSM [Kulkarni et al., CVPR 2020], BARC [Ruegg et al., CVPR 2022], BITE [Ruegg et al., CVPR 2023], VAREN [Zuffi et al., CVPR 2024], and 3D Fauna [Li et al., CVPR 2024]. Still, there may be cases where obtaining, or even defining a canonical pose of an object is challenging. Therefore, lifting the necessity of template shapes is one of the directions that we are heading to.
>
> **Limitations**
> Following NJF [Aigerman et al., ACM ToG 2022], our method uses differential operators to compute Jacobian fields. Therefore, it may not be directly applicable to meshes with defects (e.g., duplicate vertices) or multiple connected components.

---

> > ### Comment · Reviewer_skND · 2024-08-14
> >
> > Thanks for the rebuttal! Most of my concerns have been addressed, and I will raise my score. I wish the changes and comments from all reviewers could be incorporated to the camera-ready version of this paper.

---

### Author Rebuttal · Authors · 2024-08-06

We thank all reviewers for taking the time to review our submission and for providing constructive and insightful comments and feedback. We have compiled the qualitative results discussed in our rebuttal in the attached PDF file.

---

### Comment · Area_Chair_xCa4 · 2024-08-10
**Dis**

Hi reviewers,

Thank you for your hard work in reviewing the paper!
Please check out the authors' responses and ask any questions you have to help clarify things by Aug 13.

--AC

---

### Decision · Program_Chairs · 2024-09-25

**Decision:**

Accept (poster)

**Comment:**

This paper is a clear accept. It receives all positive ratings of 1 very strong accept, 1 weak accept and 1 borderline accept. The reviewers agree that the paper studies a novel and important task, the proposed method is sound and effective with clear and well-justified mathematical formulations. The first two reviewers are satisfied with the authors' rebuttal, and the last reviewer asked for some clarifications but did not point out any unsatisfaction. The authors should incorporate all the comments in the review and rebuttal into the final paper.